# A delta-tubulin/epsilon-tubulin/Ted protein complex is required for centriole architecture

**Rachel Pudlowski[1], Lingyi Xu[1], Ljiljana Milenkovic[2], Chandan Kumar[1], Katherine Hemsworth[1], Zayd Aqrabawi[1], Tim Stearns[2,3], Jennifer T Wang[1]***

[1]Department of Biology, Washington University in St. Louis, St. Louis, United States; [2]Department of Biology, Stanford University, Stanford, United States; [3]Rockefeller University, New York City, United States

*For correspondence:
wjennifer@wustl.edu

Competing interest: The authors declare that no competing interests exist.

## eLife Assessment

The study by Pudlowski et al. shows that a previously-identified protein complex, composed of delta- and epsilon-tubulin together with TEDC1 and TEDC2, functions in generating centriolar triplet microtubules, and that this is crucial for the proper formation of centriolar subdomains and the stability of centrioles throughout the cell cycle. This is an **important** study that advances our understanding of centriole biogenesis and structure and is supported by **convincing** evidence based on knockout cell lines, immunoprecipitation, and ultrastructure expansion microscopy. The work is of interest to cell biologists, in particular researchers with interest in centrosome biology.

**Abstract** Centrioles have a unique, conserved architecture formed by three linked, 'triplet', microtubules arranged in ninefold symmetry. The mechanisms by which these triplet microtubules are formed remain unclear but likely involve the noncanonical tubulins delta-tubulin and epsilon-tubulin. Previously, we found that human cells lacking delta-tubulin or epsilon-tubulin form abnormal centrioles, characterized by an absence of triplet microtubules, lack of central core protein POC5, and a futile cycle of centriole formation and disintegration (Wang et al., 2017). Here, we show that human cells lacking either TEDC1 or TEDC2 have similar abnormalities. Using ultrastructure expansion microscopy, we observed that mutant centrioles elongate to the same length as control centrioles in G2 phase and fail to recruit central core scaffold proteins. Remarkably, mutant centrioles also have an expanded proximal region. During mitosis, these mutant centrioles further elongate before fragmenting and disintegrating. All four proteins physically interact and TEDC1 and TEDC2 can form a subcomplex in the absence of the tubulins, supporting an AlphaFold Multimer model of the tetramer. TEDC1 and TEDC2 localize to centrosomes and are mutually dependent on each other and on delta-tubulin and epsilon-tubulin for localization. Our results demonstrate that delta-tubulin, epsilon-tubulin, TEDC1, and TEDC2 function together to promote robust centriole architecture, laying the foundation for future studies on the mechanisms underlying the assembly of triplet microtubules and their interactions with centriole structure.

## Introduction

The major microtubule organizing center of mammalian cells, the centrosome, is composed of two barrel-shaped centrioles surrounded by layers of pericentriolar material (reviewed in *Breslow and Holland, 2019*). The unique architecture of the centriole is highly conserved: the centriole barrel walls of approximately 250 nm in diameter by 500 nm in length are formed of compound microtubules

linked to each other through shared protofilament walls, arranged in ninefold symmetry (reviewed in *Wang and Stearns, 2017*). Centrioles exhibit proximal-distal polarity comprised of three subdomains: the proximal end with triplet microtubules, the distal end with doublet microtubules, and the central core spanning the two regions (*LeGuennec et al., 2021*). The triplet microtubules are named the A-, B-, and C-tubules. The A-tubule is a complete microtubule formed of 13 protofilaments, and the B- and C-tubules are partial tubules and share protofilament walls with adjacent tubules. The A- and B-tubules extend beyond the C-tubule to form the doublet microtubules of the centriole distal end. During ciliogenesis, the A- and B-tubules elongate further to form the ciliary axoneme (reviewed in *Wang and Stearns, 2017*).

Compound microtubules are unique to centrioles and ciliary axonemes and are conserved in almost all organisms with these organelles. Little is known about the mechanisms by which they form, or the functional roles they play within centrioles and cilia. Two non-canonical members of the tubulin super-family, delta-tubulin (TUBD1) and epsilon-tubulin (TUBE1), are required for compound microtubule formation or stability in multiple organisms (*de Loubresse et al., 2001*; *Dupuis-Williams et al., 2002*; *Dutcher and Trabuco, 1998*; *Dutcher et al., 2002*; *Gadelha et al., 2006*; *Goodenough and StClair, 1975*; *Ross et al., 2013*; *Wang et al., 2017*). Previously, we showed that human cells lacking these tubulins make aberrant centrioles that only have singlet microtubules and disintegrate in mitosis, resulting in a futile cycle of centriole formation and loss every cell cycle (*Wang et al., 2017*). These mutant centrioles fail to recruit the distal end protein POC5, indicating that compound microtubules may be required for centriole composition. We concluded that either the compound microtubules themselves, or the proteins that they associate with, are required for centriole stability through the cell cycle. Together, these results suggest that the compound microtubules may form a unique scaffold for the protein-protein interactions that define centrosomes and cilia.

The compound microtubules are directly linked to many of the substructures at the proximal, central, and distal regions within centrioles. At the proximal end, the cartwheel, a ninefold symmetric hub and spokes made from SASS6 and associated proteins, is connected to the A-tubule through the pinhead, which has been proposed to be formed of CEP135 and CPAP (*Hatzopoulos et al., 2013*; *Kraatz et al., 2016*; *Lin et al., 2013a*; *Sharma et al., 2016*). Multiple cartwheels are stacked within the centriole lumen to a height of approximately one-third of the entire centriole length (~170 nm in human centrioles; *Klena et al., 2020*). The A-tubule of one triplet is connected to the C-tubule of the adjacent triplet through a structure known as the A-C linker. Recently CCDC77, WDR67, and MIIP were identified to be components of the A-C linkers (*Bournonville et al., 2024*; *Laporte et al., 2024*). Within the central core, a helical inner scaffold imparts structural integrity upon the centriole (*Le Guennec et al., 2020*; *Steib et al., 2020*), and recruits proteins, including gamma-tubulin, to the lumen of the centriole (*Schweizer et al., 2021*). This scaffold is formed in G2-phase of the first cell cycle after centriole birth, is composed of POC5, POC1B, FAM161A, WDR90, and CCDC15 and contacts all three (A-, B-, and C-) tubules of the triplet (*Arslanhan et al., 2023*; *Laporte et al., 2024*; *Le Guennec et al., 2020*; *Steib et al., 2020*). The distal region of centrioles also has a unique protein composition, including the proteins centrin, CP110, SFI1, CEP97, CEP90, OFD1, and MNR (*Kleylein-Sohn et al., 2007*; *Kumar et al., 2021*; *Laporte et al., 2022*; *Laporte et al., 2024*; *Le Borgne et al., 2022*; *Spektor et al., 2007*). The connections between the compound microtubules and these distal end proteins are not well-understood.

Canonically, centriole formation in cycling cells is 'templated', in which one newly formed procen-triole is created at the proximal end of each pre-existing parental centriole in S-phase, resulting in four centrioles within the cell. During the first cycle after their formation, procentrioles acquire post-translational modifications, elongate, recruit the inner scaffold, lose the cartwheel, and undergo centriole-to-centrosome conversion. Additional changes occur during the second cell cycle, including acquisition of the distal and subdistal appendages that are important for ciliogenesis (*Sullenberger et al., 2020*; *Tischer et al., 2021*). Under experimental manipulations in which the parental centri-oles are ablated, centrioles can also form *de novo* in S-phase (*Wong et al., 2015*). *De novo* centriole formation can result in more than five centrioles per cell and has been shown to be error-prone (*Wang et al., 2015*), perhaps indicating differences in centriole structure or regulation. The composition and architecture of centrioles made in this manner has not been systematically characterized.

Here, we extend our original work by defining the roles of two additional proteins, TEDC1 and TEDC2, that regulate triplet microtubule formation and stability. These proteins physically interact

with TUBD1 and TUBE1 (**Breslow et al., 2018**; **Huttlin et al., 2017**; **Huttlin et al., 2021**). Loss of *Tedc1* or *Tedc2* in 3T3 cells results in a variable distribution of centriole numbers through the cell cycle, and tagged TEDC1 localizes to centrosomes (**Breslow et al., 2018**). We created *TEDC1*$^{-/-}$ or *TEDC2*$^{-/-}$ mutant cells in the same background as the *TUBD1*$^{-/-}$ and *TUBE1*$^{-/-}$ mutants and found that these cells phenocopy loss of TUBD1 or TUBE1. All four proteins interact in a complex. We find that the compound microtubules are required for recruiting the helical inner scaffold and correctly positioning the proximal end. As part of our analysis, we also determine the composition and architecture of centrioles formed *de novo* and find that these are very similar to those of procentrioles formed by templated centriole duplication. Together, these results indicate that compound microtubules are required for scaffolding substructures within centrioles and maintaining centriole stability through the cell cycle.

## Results
### Loss of TEDC1 or TEDC2 phenocopies loss of TUBD1 or TUBE1

TEDC1 and TEDC2 have been reported to physically interact with delta-tubulin and epsilon-tubulin, and loss of either *Tedc1* or *Tedc2* in 3T3 cells results in cells with a variable number of centrioles through the cell cycle (**Breslow et al., 2018**). To further dissect the phenotypes of loss of *TEDC1* or *TEDC2* and directly compare to our original report on delta-tubulin and epsilon-tubulin, we used CRISPR/Cas9 to generate strong loss of function/null mutations in *TEDC1* or *TEDC2* in the same cell type and background genotype (hTERT RPE-1 *TP53*$^{-/-}$, which will be referred to as RPE-1 *p53*$^{-/-}$) as the *TUBD1*$^{-/-}$ (delta-tubulin knockout) and *TUBE1*$^{-/-}$ (epsilon-tubulin knockout) mutant cells (**Figure 1—figure supplement 1**). By immunofluorescence staining for two centriolar proteins, centrin (CETN) and CP110, we observed that *TEDC1*$^{-/-}$ and *TEDC2*$^{-/-}$ mutant cells had similar phenotypes to each other and to *TUBD1*$^{-/-}$ and *TUBE1*$^{-/-}$ mutant cells: in an asynchronously growing culture, about half of the cells had no centrioles, and half had five or more centrioles. These phenotypes were fully rescued by expression of tagged TEDC1 (TEDC1-Halotag-3xFlag) or TEDC2 (TEDC2-V5-APEX2; **Figure 1**, **Figure 1—figure supplement 1**).

Next, we checked whether the centrioles in *TEDC1*$^{-/-}$ and *TEDC2*$^{-/-}$ mutant cells underwent a futile cycle of centriole formation and disintegration. We synchronized cells in each stage of the cell cycle, quantified the number of cells with centrioles, and found that almost all mutant cells lacked centrioles in G0/G1 phase. Centrioles formed in S-phase and disintegrated in M (**Figure 1B**). The centrioles that were present in mutant cells were immature: all centrioles were positive for the procentriole marker SASS6 and negative for the mature centriole marker CEP164 (**Figure 1C and D**). We conclude that cells lacking TEDC1 or TEDC2 also undergo a futile cycle, similar to cells lacking delta-tubulin or epsilon-tubulin (**Figure 1G**).

We also examined the centriolar microtubule status of *TEDC1*$^{-/-}$ and *TEDC2*$^{-/-}$ mutant cells by TEM. Similar to cells lacking delta-tubulin or epsilon-tubulin, we found that centrioles in *TEDC1*$^{-/-}$ and *TEDC2*$^{-/-}$ mutant cells lacked compound microtubules and only had singlet microtubules. These centrioles had cartwheels and pinheads, but A-C linkers were not visible (**Figure 1E and F**, **Figure 1—figure supplement 2**). Together, these results demonstrate that loss of TEDC1 or TEDC2 phenocopies loss of delta-tubulin or epsilon-tubulin, indicating that these proteins likely act together.

### TEDC1 and TEDC2 localize to centrosomes

Next, we investigated the localization of TEDC1 and TEDC2 to determine if they may directly act on centrosomes. TEDC1 and TEDC2 are expressed at low levels in cells (**Figure 1—figure supplement 1**), and we could not reproducibly localize the endogenous proteins with antibody staining. Instead, we localized the functional, tagged proteins in our rescue cell lines. We found that the tagged rescue constructs localize to centrosomes (**Figure 2A and B**) and the antibodies for the tags were specific (**Figure 2—figure supplement 1E-J**). TEDC1 and TEDC2 were enriched at centrosomes in S/G2 and colocalized with SASS6, but not centrin, indicating that TEDC1 and TEDC2 may localize to newly formed procentrioles and/or the proximal ends of parental centrioles.

To analyze TEDC1 and TEDC2 localization at higher resolution, we localized our tagged rescue constructs using three methods: a super-resolution spinning disk confocal microscope with immunofluorescence microscopy (**Figure 2—figure supplement 1A, B**), ultrastructure expansion microscopy

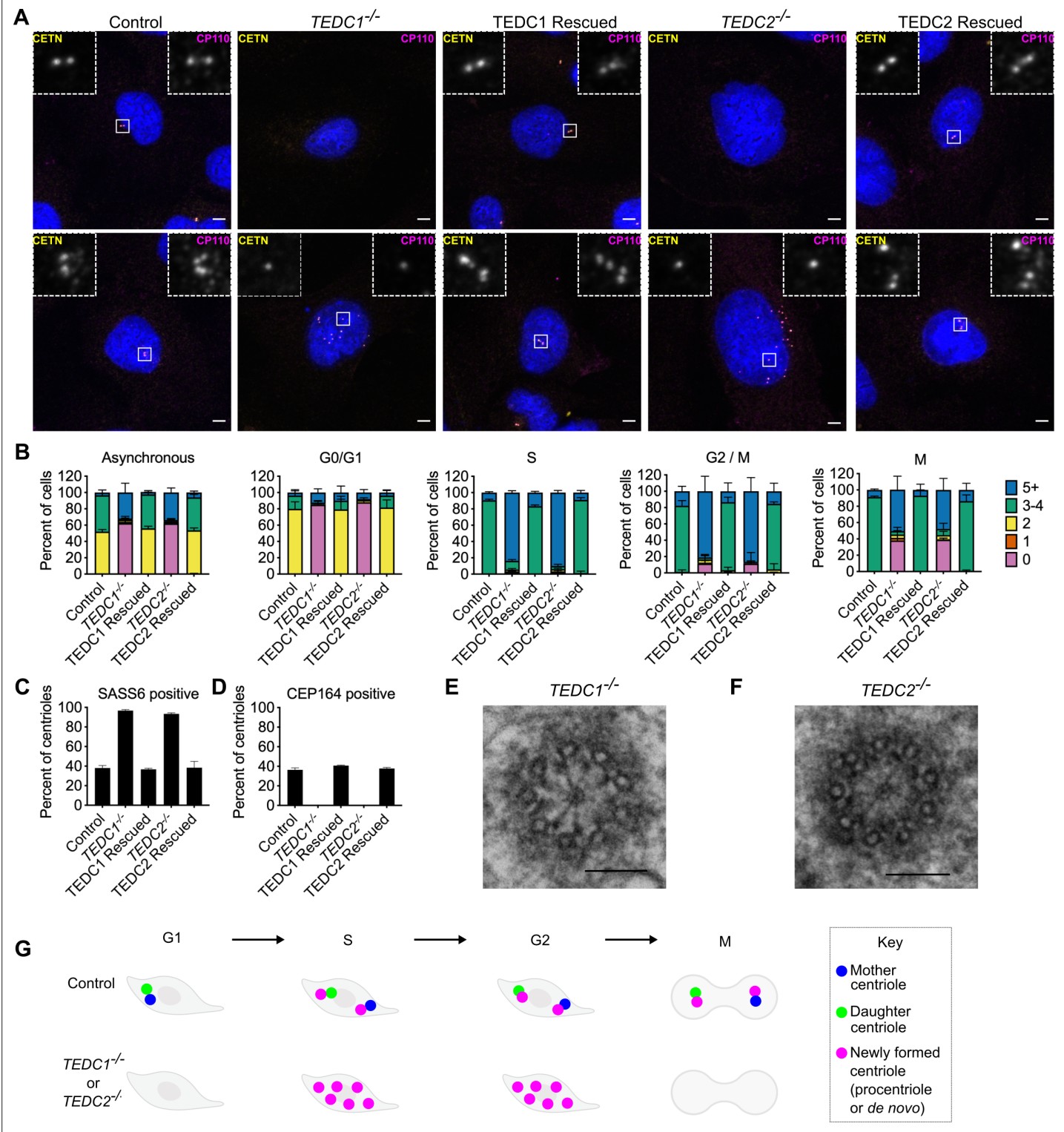

**Figure 1.** Loss of TEDC1 or TEDC2 phenocopies loss of delta-tubulin or epsilon-tubulin. (**A**) Immunofluorescence staining of control (RPE1 *TP53*$^{-/-}$), *TEDC1*$^{-/-}$ (RPE1 *TP53*$^{-/-}$; *TEDC1*$^{-/-}$), TEDC1 Rescued (RPE1 *TP53*$^{-/-}$; *TEDC1*$^{-/-}$; TEDC1-Halotag-3xflag), *TEDC2*$^{-/-}$ (RPE1 *TP53*$^{-/-}$; *TEDC2*$^{-/-}$), TEDC2 Rescued (RPE1 *TP53*$^{-/-}$; *TEDC2*$^{-/-}$; TEDC2-V5-APEX2) cells. Top row: G1 stage cells with 2 centrioles. Bottom row: S/G2 stage cells with 4 centrioles. Blue: DAPI; Yellow: Centrin (CETN); Magenta: CP110. Images are maximum projections of confocal stacks. Scale bar: 5 µm (**B**) Centriole number counts of the indicated cell lines. Cells were either asynchronous, serum-starved for G0/G1, stained for PCNA for S-phase, synchronized with RO-3306 for G2/M, or mitotic figures were identified by DAPI staining. Each condition was performed in triplicate, with n=100 cells scored for each. (**C**) Percent of all centrioles

*Figure 1 continued*

(parental, pro-, and *de novo* centrioles) in indicated cell types positive for SASS6 staining. Each condition was performed in triplicate, with 200 cells scored for each. (**D**) Percent of all centrioles (parental, pro-, and *de novo* centrioles) in indicated cell types positive for CEP164 staining. Each condition was performed in triplicate, with 100 cells scored for each. (**E**) TEM cross-section of a centriole in a G2-phase *TEDC1⁻/⁻* cell. Scale bar: 100 nm (**F**) TEM cross-section of a centriole in a G2-phase *TEDC2⁻/⁻* cell. Scale bar: 100 nm (**G**) Schematic of centriole formation and loss in control and *TEDC1⁻/⁻* or *TEDC2⁻/⁻* cells.

The online version of this article includes the following source data and figure supplement(s) for figure 1:

**Source data 1.** Raw data (centriole counts) for *Figure 1B, C and D*.

**Figure supplement 1.** Creation of *TEDC1⁻/⁻* and *TEDC2⁻/⁻* mutant cell lines.

**Figure supplement 1—source data 1.** Original files of full uncropped, unedited blots in *Figure 1—figure supplement 1E and F*.

**Figure supplement 1—source data 2.** Labeled uncropped, unedited blots in *Figure 1—figure supplement 1E and F*.

**Figure supplement 2.** Symmetrization of *TEDC1⁻/⁻* and *TEDC2⁻/⁻* mutant centrioles.

---

(U-ExM, (*Gambarotto et al., 2019*), *Figure 2C and D*), and a second expansion microscopy method (*Kong et al., 2024*, *Figure 2—figure supplement 1C, D*). With all three methods, we observed that both proteins localize to procentrioles and the proximal ends of parental centrioles. At these regions, both proteins overlap with the centriolar microtubules. Together, these results show that TEDC1 and TEDC2 localize to centrosomes and likely directly act upon them.

## TEDC1, TEDC2, TUBD1, and TUBE1 form a complex in cells

To determine how TEDC1, TEDC2, TUBD1 and TUBE1 might act together, we first determined whether they are mutually required for their localization at centrosomes. We found that TEDC1 did not localize to centrioles in the absence of TEDC2, TUBD1, or TUBE1 (*Figure 3A*). Likewise, TEDC2 did not localize to centrioles in the absence of TEDC1, TUBD1, or TUBE1 (*Figure 3B*). These results indicate that these proteins are mutually required for TEDC1 or TEDC2 localization. Furthermore, overexpression of TEDC1 or TEDC2 did not rescue the centriole phenotypes in any of the other mutants, indicating that TEDC1 and TEDC2 are not downstream effectors of TUBD1 and TUBE1 (*Figure 3A and B*).

TEDC1 and TEDC2 have previously been shown to physically interact with TUBD1 and TUBE1 (*Breslow et al., 2018*). To further probe the nature of this interaction, we first determined whether any of these proteins may form subcomplexes in cells. We expressed TEDC1-Halotag-3xFlag in each mutant cell line and determined whether immunoprecipitation of tagged TEDC1 could precipitate the other proteins. TEDC1-Halotag-3xFlag rescuing the *TEDC1⁻/⁻* mutant could precipitate TEDC2, TUBD1, and TUBE1, indicating that all four proteins physically interact. TEDC1 did not interact with epsilon-tubulin in the absence of delta-tubulin, nor did it interact with delta-tubulin in the absence of TUBE1. In the absence of TEDC2, TEDC1 did not interact with TUBD1 or TUBE1. However, in the absence of TUBD1 or TUBE1, TEDC1 and TEDC2 could still interact with each other (*Figure 3C*).

We performed the reciprocal experiment, in which we expressed TEDC2-V5-APEX2 in each mutant cell line and determined whether immunoprecipitation of tagged TEDC2 could precipitate the other proteins. We observed similar results as our analysis with TEDC1. TEDC2-V5-APEX2 rescuing the *TEDC2⁻/⁻* mutant could precipitate TEDC1, TUBD1, and TUBE1, indicating that all four proteins physically interact. TEDC2 did not interact with either tubulin in the absence of the other. In the absence of TEDC1, TEDC2 did not interact with either tubulin. However, in the absence of TUBD1 or TUBE1, TEDC2 and TEDC1 could still interact (*Figure 3D*).

Together, these experiments indicate that TEDC1, TEDC2, TUBD1, and TUBE1 physically interact with each other, as previously reported (*Breslow et al., 2018*; *Huttlin et al., 2017*; *Huttlin et al., 2021*). Furthermore, TEDC1 and TEDC2 can form a subcomplex in the absence of either tubulin.

To gain additional insight into the nature of this interaction, we used AlphaFold-Multimer (*Evans et al., 2021*) to predict the structure of the complex. AlphaFold-Multimer predicted that TUBD1 and TUBE1 would form a heterodimer, similar to the alpha-tubulin/beta-tubulin heterodimer, with TUBD1 at the minus-end of the heterodimer. AlphaFold also predicted that the alpha-helices of TEDC1 and TEDC2 interact with each other, and that TEDC1 and TEDC2 form an interaction surface with TUBD1. These predictions, especially at the interface between TEDC1, TEDC2, and TUBD1, yielded high confidence pLDDT and PAE scores (*Figure 3E-G*, *Figure 3—figure supplement 1A*). A similar prediction

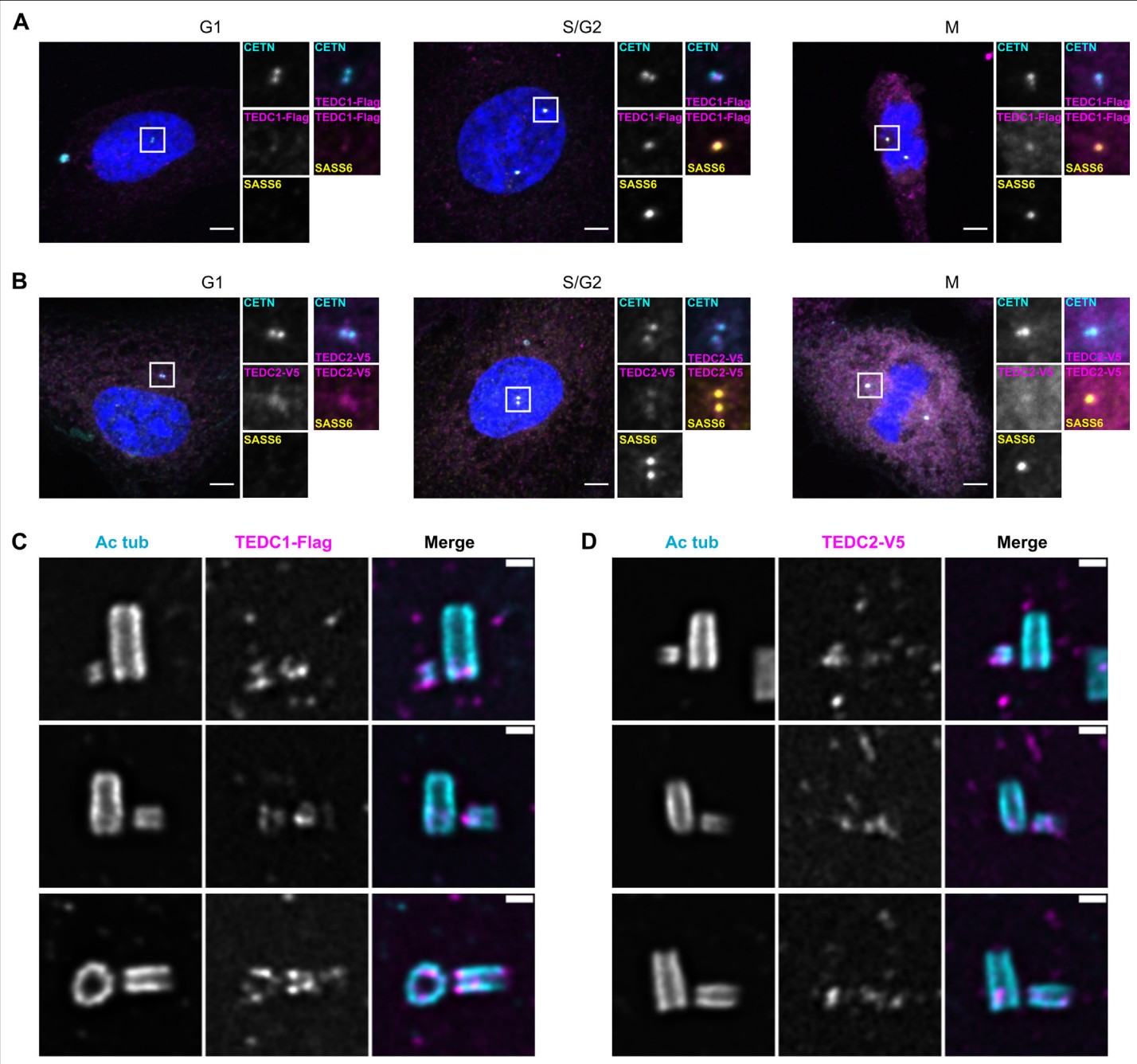

**Figure 2.** TEDC1 and TEDC2 localize to centrioles. (**A**) Immunofluorescence staining of TEDC1 rescue cell lines expressing TEDC1-Halotag-3xFlag in G1, S/G2, and M. Images are maximum projections of confocal stacks. Blue: DAPI; Cyan: Centrin (CETN); Magenta: TEDC1-Halotag-3xFlag (localized with anti-Flag antibody); Yellow: SASS6. Scale bar: 5 µm. (**B**) Immunofluorescence staining of TEDC2 rescue cell lines expressing TEDC2-V5-APEX2 in G1, S/G2, and M. Images are maximum projections of confocal stacks. Blue: DAPI; Cyan: Centrin (CETN, localized with anti-GFP antibody recognizing GFP-centrin); Magenta: TEDC2-V5-APEX2 (localized with anti-V5 antibody); Yellow: SASS6. Scale bar: 5 µm. (**C**) U-ExM of TEDC1 rescue cell lines expressing TEDC1-Halotag-3xFlag, arranged by procentriole length. Cyan: Acetylated tubulin; Magenta: TEDC1-Halotag-3xFlag (localized with anti-Flag antibody). Confocal image stacks were deconvolved using Microvolution; single plane images shown. Scale bar: 1 µm. (**D**) U-ExM of TEDC2 rescue cell lines expressing TEDC2-V5-APEX2, arranged by procentriole length. Cyan: Acetylated tubulin; Magenta: TEDC2-V5-APEX2 (localized with anti-V5 antibody). Confocal image stacks were acquired with a Yokogawa CSU-W1 spinning disk microscope and deconvolved using Microvolution; single plane images shown. Scale bar: 1 µm.

The online version of this article includes the following figure supplement(s) for figure 2:

**Figure supplement 1.** Extended localization analyses of TEDC1 and TEDC2.

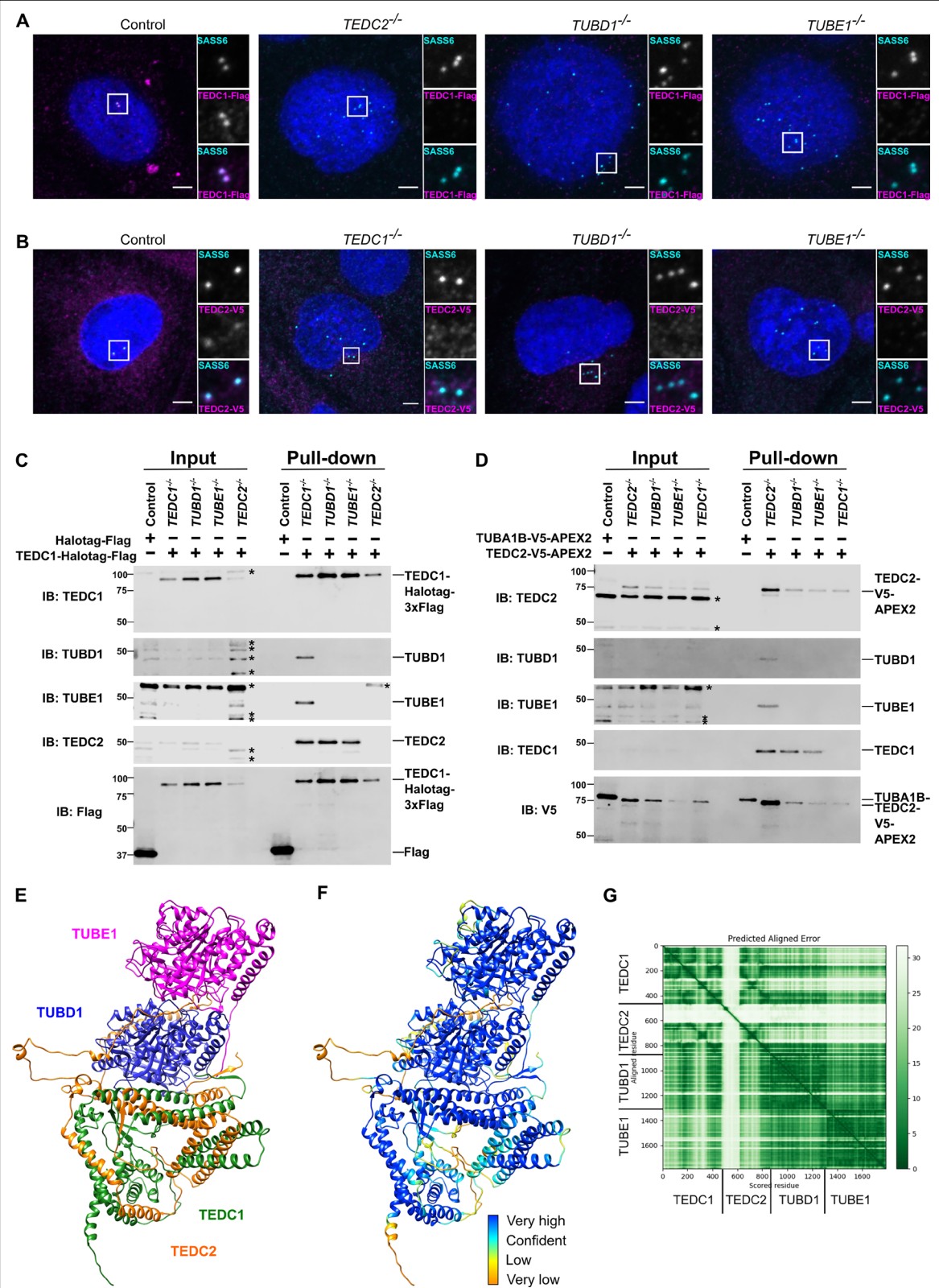

**Figure 3.** TEDC1, TEDC2, TUBD1, TUBE1 form a complex in cells. (**A**) Centrosomal TEDC1 localization depends on TEDC2, TUBD1, TUBE1. Immunofluorescence staining of cells expressing TEDC1-Halotag-3xFlag. Control cell is *TEDC1⁻/⁻* mutant cells rescued with TEDC1-Halotag-3xFlag. Images are maximum projections of confocal stacks. Blue: DAPI; Cyan: SASS6; Magenta: TEDC1-Halotag-3xFlag (localized with anti-Flag antibody). Scale bar: 5 μm. (**B**) Centrosomal TEDC2 localization depends on TEDC1, TUBD1, TUBE1. Immunofluorescence staining of cells expressing TEDC2-

*Figure 3 continued*

V5-APEX2. Control cell is TEDC2 mutant cells rescued with TEDC2-V5-APEX2. Images are maximum projections of confocal stacks. Blue: DAPI; Cyan: SASS6; Magenta: TEDC2-V5-APEX2 (localized with anti-V5 antibody). Scale bar: 5 µm. (C) TEDC1 pulls down TEDC2 in the absence of delta or epsilon-tubulin. Western blot of input and pulldown of Halotag-Flag or TEDC2-Halotag-Flag in the indicated cell lines. Control cells are *TP53*⁻/⁻ cells expressing Halotag-3xFlag. IB: indicates the antibody used for immunoblotting. The proteins and their positions are labeled on the right. Asterisks mark non-specific bands. (D) TEDC2 pulls down TEDC1 in the absence of delta or epsilon-tubulin. Western blot of input and pulldown of TUBA1B-V5-APEX2 or TEDC2-V5-APEX2 in the indicated cell lines. Control cells are *TP53*⁻/⁻ cells expressing TUBA1B-V5-APEX2. IB: indicates the antibody used for immunoblotting. The proteins and their positions are labeled on the right. Asterisks mark non-specific bands. (E) AlphaFold-Multimer prediction of the complex (F) AlphaFold-Multimer prediction colored according to pLDDT. Very high: pLDDT > 90. High: 90 > pLDDT > 70. Low: 70 > pLDDT > 50. Very low: pLDDT <50 (G) Predicted align error of the AlphaFold Multimer prediction. Expected position error (Angstroms) is graphed.

The online version of this article includes the following source data and figure supplement(s) for figure 3:

**Source data 1.** Original files of full uncropped, unedited blots in *Figure 3C and D*.

**Source data 2.** Labeled uncropped, unedited blots in *Figure 3C and D*.

**Figure supplement 1.** AlphaFold-Multimer and AlphaFold3 predictions.

was obtained with the newly released AlphaFold 3 (*Abramson et al., 2024*; *Figure 3—figure supplement 1B*). As controls, we used AlphaFold-Multimer to predict whether TEDC1 and TEDC2 might interact with alpha-tubulin and beta-tubulin, and whether similar structures would be predicted for *Xenopus* TEDC1, TEDC2, TUBD1, and TUBE1. While AlphaFold-Multimer did not predict a high-confidence interaction for TEDC1, TEDC2, alpha- and beta-tubulin (*Figure 3—figure supplement 1C*), it did predict a high-confidence structure for *Xenopus* TEDC1, TEDC2, TUBD1, and TUBE1, similar to that predicted for the human proteins (*Figure 3—figure supplement 1D*).

Our pulldown experiments showed that TEDC1 and TEDC2 can interact in a subcomplex in the absence of TUBD1 or TUBE1, which supports the predicted structural model, in which TEDC1 and TEDC2 are predicted to directly interact with each other without being bridged by either tubulin. Further supporting this model, immunoprecipitation of TEDC2 identifies the other proteins in stoichiometric amounts (*Breslow et al., 2018*), and we previously showed that TUBD1 and TUBE1 physically interact (*Wang et al., 2017*). Given the size and shape of the tetrameric complex as predicted by AlphaFold-Multimer, it is possible that these may form a structural component of centrioles. Future work will be necessary to test these possibilities. Together, our experiments indicate that TEDC1, TEDC2, TUBD1, and TUBE1 physically interact in a complex and are recruited together to centrioles.

## Loss of TEDC1, TEDC2, TUBD1, or TUBE1 results in centrioles with aberrant ultrastructure

Next, we determined how the loss of these proteins, and the triplet microtubules themselves, affect centriole ultrastructure and protein composition. Because centrioles are constitutively formed *de novo* every cell cycle in our mutant cells, we incorporated two controls in our analysis: procentrioles undergoing normal parental-mediated centriole duplication in control (RPE-1 *p53*⁻/⁻) cells, and centrioles formed in RPE-1 *p53*⁻/⁻ cells *de novo* in the first cell cycle after centrinone washout (*Wong et al., 2015*). For each of the two control and four mutant cell lines, cells were synchronized by mitotic shake off, resulting in coverslips enriched for cells in late S- and G2-phases, with a minor population in M-phase. Synchronized cells were then expanded using U-ExM and stained for centriolar markers.

We first tested whether the microtubules of mutant centrioles could be modified by acetylation of alpha-tubulin. During centriole formation, acetylation is thought to proceed from the proximal toward the distal end and from the A- to the C-tubules (*Sahabandu et al., 2019*). We found that antibodies against acetylated alpha-tubulin stained mutant centrioles well (*Figure 4B*), indicating that centrioles with only singlet A-tubules can be acetylated.

We next tested whether mutant centrioles were capable of elongating during the cell cycle. In our expansion gels of cells enriched in late S and G2 phases, we used PCNA to mark S-phase cells and co-stained with acetylated tubulin to mark centrioles. Similar to a recently published report, we also found a range of centriole lengths in S- and G2-phases (*Laporte et al., 2024*). In S-phase, centrioles were short in all conditions. In G2-phase, centrioles elongated in all conditions, and mutant centrioles reached approximately similar lengths as control centrioles (*Figure 4A*). By contrast, mutant centriole widths did not increase and centrioles remained narrow, as we previously reported (*Figure 4—figure supplement 5* and *Wang et al., 2017*). These results indicate that centrioles with singlet microtubules

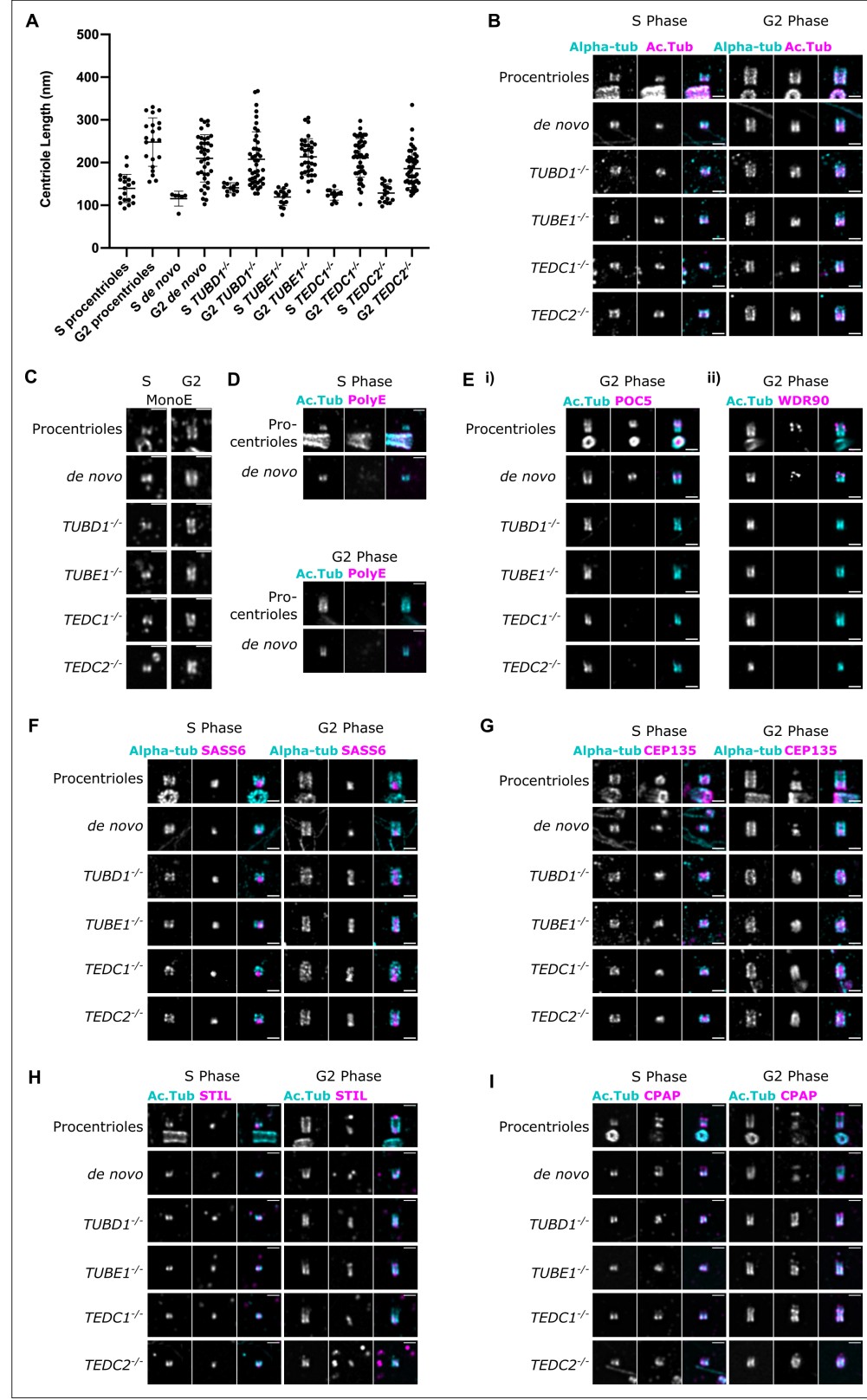

**Figure 4.** Mutant centrioles elongate in G2 but fail to recruit central core proteins and have an expanded proximal region. (**A**) Lengths of expanded centrioles from cells of the indicated cell cycle stages. Lengths were adjusted for the gel expansion factors. Cells were synchronized in S/G2/M and S-phase cells were marked with PCNA. For each genotype, the differences between S and G2 phase centriole lengths are statistically significant (*p*<0.0001, Welch's

*Figure 4 continued on next page*

*Figure 4 continued*

t-test). (**B**) U-ExM images of centrioles stained for alpha-tubulin and acetylated tubulin. (**C**) U-ExM of centrioles in S or G2 phase stained with monoE (GT335) antibody. (**D**) U-ExM of control centrioles in S or G2 phase stained with acetylated tubulin and polyE antibodies (**Ei**) U-ExM of centrioles in G2-phase stained with acetylated tubulin (cyan) and POC5 (magenta) antibodies. POC5 is present in the central core of control procentrioles and *de novo* centrioles and absent from mutants. (**Eii**) U-ExM of centrioles in G2 phase stained with acetylated tubulin (cyan) and WDR90 (magenta) antibodies. WDR90 is present in the central core of control procentrioles and *de novo* centrioles, and absent from mutants. (**F–I**) U-ExM of centrioles in S and G2 phase stained for alpha tubulin (cyan) or acetylated tubulin (Ac Tub, cyan) and the following antibodies in magenta: (**F**) SASS6, (**G**) CEP135, (**H**) STIL, (**I**) CPAP. In control centrioles, these proteins are limited to the proximal end. In mutant centrioles, these proteins are present at the proximal end in S phase centrioles and elongate throughout the entire centriole in G2 phase. Images were acquired with a Yokogawa CSU-W1 SoRA with 2.8 x relay and deconvolved with 10 iterations using Microvolution. Scale bars: 1 µm.

The online version of this article includes the following source data and figure supplement(s) for figure 4:

**Source data 1.** Data for *Figure 4A*.

**Figure supplement 1.** Extended analyses of mutant centriole architecture and U-ExM gel expansion factor.

**Figure supplement 1—source data 1.** Data for *Figure 4—figure supplement 1G*.

**Figure supplement 2.** Total protein levels of centrosomal proteins are unchanged in mutant cells.

**Figure supplement 2—source data 1.** Original files of full uncropped, unedited blots in *Figure 4—figure supplement 2*.

**Figure supplement 2—source data 2.** Labeled uncropped, unedited blots in *Figure 4—figure supplement 2*.

**Figure supplement 3.** Quantification of CEP135 centriolar localization through S and G2 phases.

**Figure supplement 3—source data 1.** Data for *Figure 4—figure supplement 3*.

**Figure supplement 4.** Quantification of SASS6 centriolar localization through S and G2 phase.

**Figure supplement 4—source data 1.** Data for *Figure 4—figure supplement 4*.

**Figure supplement 5.** Quantification of centriole widths and lengths.

**Figure supplement 5—source data 1.** Data for *Figure 4—figure supplement 5*.

can elongate to the same overall length as control centrioles in G2 phase. Consistent with this hypothesis, CEP120, a protein involved in regulating centriole length (*Comartin et al., 2013*; *Lin et al., 2013b*; *Mahjoub et al., 2010*), was present and properly localized within mutant centrioles (*Figure 4—figure supplement 1D*).

The compound microtubules of centrioles are heavily post-translationally modified, and recent studies have indicated that each tubule may acquire different modifications (*Guichard et al., 2023*). We checked glutamylation, a post-translational modification thought to be restricted to the outer surface of centrioles (*Guichard et al., 2023*). Within *Chlamydomonas* centrioles, glutamylation is differentially distributed between each tubule: on the C-tubule at the distal end, on all three tubules in the central core, and on the A-tubule at the proximal end (*Hamel et al., 2017*). In human centrioles, polyglutamylation is enriched in the proximal and central regions, and is absent in the distal region (*Gambarotto et al., 2019*; *Mahecic et al., 2020*; *Sullenberger et al., 2020*). We used two antibodies to detect glutamylation: the GT335 antibody, which recognizes the glutamylation branch and thus detects all polyglutamylation, and the polyE antibody, which recognizes long polyglutamate side chains with at least 2 or 3 glutamate residues (*Kann et al., 2003*; *van Dijk et al., 2007*). We found that mutant and control centrioles could be stained by GT335 (*Figure 4C*), indicating that mutant centrioles are at least mono-glutamylated. However, the polyE antibody did not label control procentrioles or *de novo* centrioles in the first cell cycle after their formation, making this antibody uninformative for our mutants (*Figure 4D*). These results show that centrioles with just singlet microtubules (A-tubules) can be mono-glutamylated. Moreover, similar to previous reports (*Sullenberger et al., 2020*), our results suggest that centriole glutamylation is a multi-step process, in which long glutamate side chains are added later during centriole maturation.

We previously demonstrated that *TUBD1^-/-* and *TUBE1^-/-* mutant centrioles fail to recruit the distal centriole protein POC5 (*Wang et al., 2017*). Using expansion microscopy, we found that *TEDC1^-/-* and *TEDC2^-/-* mutant centrioles also failed to recruit POC5 (*Figure 4Ei*). Since our original work was published, POC5 was shown to be a component of the helical inner scaffold within the central core.

These results indicate that the helical inner scaffold is not properly formed in centrioles with singlet microtubules. To test the mechanisms underlying loss of POC5, we next tested whether mutant centrioles recruit WDR90, which has been proposed to localize to the inner junction between the A- and B-tubules and function in recruiting the inner scaffold (*Steib et al., 2020*). We found that WDR90 was not recruited to mutant centrioles, in contrast to control centrioles, in which it is recruited in G2-phase (*Figure 4Eii*). From these results, it is likely that mutant centrioles with singlet microtubules fail to build or stabilize the inner junction between the A- and B-tubules. In the absence of the inner junction and junctional protein WDR90, centrioles with singlet microtubules cannot form the inner scaffold. As also previously reported (*Laporte et al., 2024*), we failed to detect gamma-tubulin within the lumen of control or *de novo*-formed centrioles in S or G2-phase (*Figure 4—figure supplement 1E*) and thus were unable to test whether gamma-tubulin, which is recruited to the lumen of centrioles by the inner scaffold, was mislocalized in mutant centrioles.

Next, we tested whether the centriole proximal end might be properly formed in mutant centrioles. We found that the centriolar cartwheel protein, SASS6, was present within the lumen of control and mutant centrioles in S-phase. In control centrioles in G2-phase, SASS6 was restricted to just the proximal end. Surprisingly, SASS6 was elongated in all G2-phase mutant centrioles (*Figure 4F*, *Figure 4—figure supplement 4F*). We observed a similar phenotype with multiple other proximal-end proteins: CEP135, STIL, CPAP, and CEP44 (*Figure 4G-I*, *Figure 4—figure supplement 1*, *Figure 4—figure supplement 3*), indicating that the entire proximal end is elongated in mutant centrioles. The extended localization of proximal end proteins was not due to increased protein expression in mutant cells (*Figure 4—figure supplement 2*). We conclude that loss of TEDC1, TEDC2, TUBD1, or TUBE1 results in elongated proximal end domains within mutant centrioles.

Elongation of the proximal end of centrioles may also indicate an overall defect in centriole polarity. To test this hypothesis, we next determined whether these mutant centrioles might properly recruit proteins to their distal ends. We found that CETN2 and CP110, two proteins of the distal centriole, were localized to mutant centrioles and clearly marked one end of the centriole barrel in both S-phase and G2-phase (*Figure 4—figure supplement 1B, C*). We conclude that proximal-to-distal centriole polarity was unaffected in mutant centrioles, and proximal end elongation did not affect the recruitment of proteins to the centriole distal end. Together, these results indicate that centrioles lacking compound microtubules are unable to properly regulate the length of the proximal end.

## Mutant centrioles elongate further in mitosis before fragmenting

Centrioles lacking triplet microtubules undergo a futile cycle of formation and disassembly, but the mechanisms underlying disassembly are not well-understood. We first tested whether centriole loss in mutant centrioles may be due to loss of CEP295. CEP295 promotes centriole-to-centrosome conversion, a process in which pericentriolar material is recruited to newly-formed procentrioles. Cells lacking CEP295 form centrioles that disintegrate during the cell cycle due to a failure to undergo centriole-to-centrosome conversion (*Izquierdo et al., 2014*). Using U-ExM, we found that CEP295 was present and normally localized within mutant centrioles in both S- and G2-phases (*Figure 4—figure supplement 1F*). We conclude that centriole loss in our mutants is unlikely to be due to loss of CEP295 localization, and therefore that TEDC1, TEDC2, TUBD1, and TUBE1 are likely part of a different pathway required for centriole stability through the cell cycle.

Next, we used U-ExM to visualize centriole loss during mitosis. We stained for the centriole wall (GT335), the centriole proximal end (SASS6) and the centriole distal end (CP110). In control cells, in which centrioles formed *de novo* after centrinone washout, multiple centrioles could be seen throughout mitosis, and SASS6 was lost from centrioles in anaphase-stage cells (*Figure 5A and B*). By contrast, in prometaphase stage *TUBD1*[-/-] or *TUBE1*[-/-] cells, we found that centrioles had a unique appearance: they were longer than normal, with an elongated proximal end marked by SASS6, and a CP110-positive cap. These two ends were connected by weak monoE staining (*Figure 5C and E*). This phenotype is identical to our observations of centrioles in a prometaphase *TUBE1*[-/-] cell by TEM in our previous publication (Figure 2B in *Wang et al., 2017*). After metaphase, centrioles in mutant cells were either completely absent, or had a fragmented appearance (*Figure 5D and F*), with aggregates of staining that did not resemble true centrioles. We conclude that in our mutant cells, centrioles elongate in early mitosis to form an aberrant intermediate structure, followed by fragmentation in late mitosis.

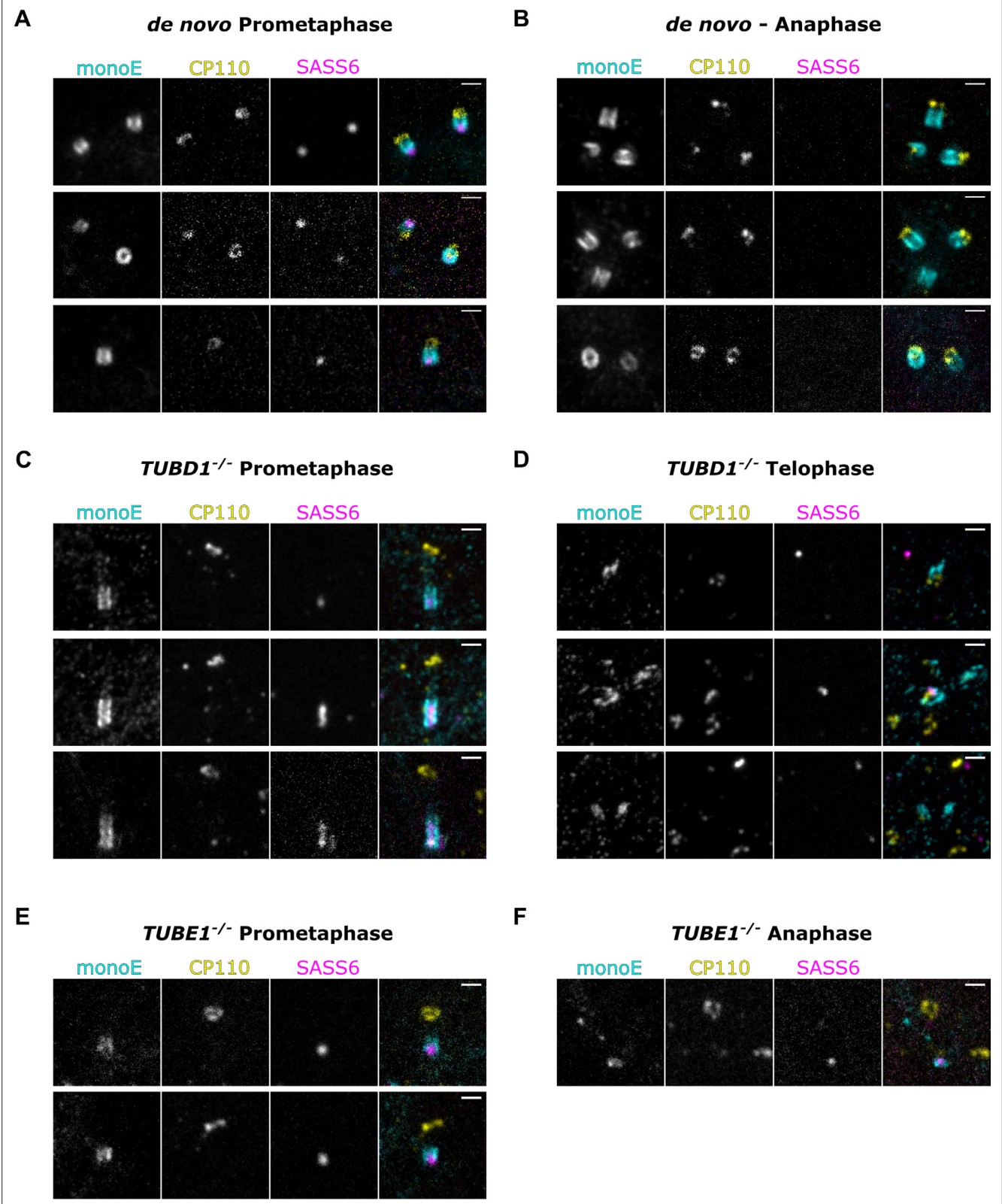

**Figure 5.** Mutant centrioles elongate further in mitosis before fragmenting. U-ExM of centrioles stained for monoE (GT335, cyan), CP110 (yellow) and SASS6 (magenta). (**A**) A prometaphase cell with centrioles formed *de novo* after centrinone washout (**B**) An anaphase cell with centrioles formed *de novo* (**C**) A prometaphase *TUBD1⁻/⁻* cell (**D**) A telophase *TUBD1⁻/⁻* cell (**E**) A prometaphase *TUBE1⁻/⁻* cell (**F**) An anaphase *TUBE1⁻/⁻* cell. Scale bars: 1 um. Images were acquired with a Yokogawa CSU-W1 SoRA with 2.8 x relay.

## Discussion

Here, we extend our previous study on delta-tubulin (TUBD1), epsilon-tubulin (TUBE1) and the centriolar triplet microtubules. Previously, we showed that loss of either of these proteins from mammalian cultured cell lines results in the same phenotype: loss of the triplet microtubules and a futile cycle of centriole formation and disintegration (*Wang et al., 2017*). Here, we add two new proteins to this pathway: TEDC1 and TEDC2, which were originally identified by their association with TUBD1 and TUBE1 (*Breslow et al., 2018*; *Huttlin et al., 2017*; *Huttlin et al., 2021*). Loss of TEDC1 or TEDC2 phenocopies the loss of TUBD1 or TUBE1: aberrant centrioles are formed that lack triplet microtubules and disintegrate during passage through mitosis. TEDC1 and TEDC2 localize to centrioles, indicating that they have a direct role in forming or maintaining centriole structure, and their localization depends on each of the other three proteins within the complex. All four proteins physically interact with each other. Using our mutant cell lines, we interrogated whether any of these proteins can form subcomplexes within cells. We found that TEDC1 and TEDC2 can interact with each other independently of the tubulins, supporting a predicted AlphaFold-Multimer model. Together, these results indicate that these four proteins act together in a complex at centrosomes to form or stabilize the compound microtubules.

While the molecular mechanisms underlying the function of this complex are unknown, an attractive model is that the tetrameric complex forms a structural component of centrioles. Our AlphaFold models indicate that such a structure would be approximately 13 nm in length and 6 nm in width. Within procentrioles and the proximal region of the parental centriole, it is possible that these four proteins help form the A-C linker, the pinhead, or the triplet base. Recently, components of the A-C linker have been identified (*Bournonville et al., 2024*; *Laporte et al., 2024*), and three of the proteins in our complex (TEDC2, TUBD1, and TUBE1) had shared co-dependencies with A-C linker components using DepMap (*Bournonville et al., 2024*). The A-C linker is lost from our mutant centrioles, but it is not clear whether this is because these proteins have a direct role in forming A-C linkers or whether this reflects an indirect role of the triplet microtubules in stabilizing A-C linkers. We note that it is also possible that only some proteins of the complex, such as delta-tubulin and epsilon-tubulin, form structural components of centrioles, or that the complex may interact transiently with centrioles. Future experiments will reveal the mechanisms by which these proteins act.

Using ultrastructure expansion microscopy, we find that mutant centrioles with singlet microtubules exhibit additional major architectural defects, including absence of the inner scaffold and elongation of the proximal end. We propose that the absence of the inner scaffold arises from the loss of the B- and C-tubules within centrioles, which may serve to anchor WDR90 and/or other proteins of the inner scaffold. WDR90 has been proposed to localize to the inner junction between the A- and B-tubules and is required for recruiting other inner scaffold components (*Le Guennec et al., 2020*; *Steib et al., 2020*). We find that mutant centrioles with singlet microtubules fail to localize WDR90, and thus speculate that the B-tubule is required to recruit or stabilize WDR90 at the inner junction. In addition, by cryo-electron tomography, the inner scaffold makes connections to all three (A-, B-, and C-) tubules. Although the identities of all the proteins that form these connections have not been determined, it is possible that mutant centrioles with only A-tubules also fail to provide anchoring sites for the other proteins within the inner scaffold. Together, these results demonstrate that the compound microtubules of centrioles are required for proper formation of the inner helical scaffold of the central core.

Mutant centrioles with singlet microtubules have an elongated proximal end that extends the entire length of the centriole, as marked by multiple proximal end markers (SASS6, CEP135, STIL, CPAP, CEP44). These results are also supported by our previous observations that by TEM, the lumen of *TUBD1*[-/-] and *TUBE1*[-/-] mutant centrioles are filled with electron-dense material (*Wang et al., 2017*). Little is known about the molecular mechanisms that regulate proximal end length, though centrioles from the symbiotic flagellate *Trichonympha* bear an elongated proximal region with extended cartwheel, and the doublet and singlet-bearing centrioles from *Drosophila* and *C. elegans* have cartwheels that extend the entire length of the centriole (*González et al., 1998*; *Guichard and Gönczy, 2016*; *Guichard et al., 2012*; *Pelletier et al., 2006*; *Woglar et al., 2022*). It is possible that the triplet microtubules, the inner scaffold, and/or the TUBD1/TUBE1/TEDC1/TEDC2 protein complex might act to limit the length of the proximal end. Recently, loss of the inner scaffold protein POC1A has been shown to result in centrioles with extended regions of some proximal proteins, including CEP44, CEP135, and CEP295, indicating that the inner scaffold regulates the extent of these proteins (*Sala*

*et al., 2024*). Interestingly, unlike our mutant centrioles which have singlet microtubules, *POC1A*[-/-] mutant centrioles can form triplet microtubules and do not have extended SASS6 staining (*Sala et al., 2024*). This suggests that the height of the cartwheel may be regulated by the triplet microtubules. The cartwheel and centriolar microtubules have been proposed to assemble interdependently to impart ninefold symmetry upon the centriole (*Hilbert et al., 2016*), and it is possible that interdependent assembly also regulates the height of the cartwheel.

Many aspects of centriole architecture, including formation of the distal tip, centriole length regulation prior to mitosis, acquisition of post-translational modifications, establishment of proximal-distal polarity, and recruitment of proteins required for centriole-to-centrosome conversion, are unaffected in mutant centrioles. These results indicate that the proteins that regulate these processes can act upon the A-tubule independently of the B- and C-tubules.

Here, we also extend our previous observations of centriole loss in mutant centrioles. In most cell types, centrioles are inherited by daughter cells during each mitosis. Centriole loss is not unique to centrioles lacking compound microtubules: mammalian cells engineered to lack CEP295 also form centrioles that are lost through the cell cycle, due to an inability to undergo centriole to centrosome conversion (*Izquierdo et al., 2014*). Similarly, in *Drosophila* oocytes, down-regulation of Polo kinase and pericentriolar material triggers centriole elimination (*Pimenta-Marques et al., 2016*). We find that CEP295 is properly localized in mutant centrioles with singlet microtubules, indicating that centriole loss in this context may be independent of centriole to centrosome conversion and pericentriolar material recruitment. Using expansion microscopy, we find that centriole loss is correlated with loss of the SASS6 cartwheel in mitosis. In this regard, mutant centrioles with singlet microtubules resemble centriole loss within *C. elegans* oocytes, in which an analogous structure to the cartwheel named the central tube is lost prior to centriole widening and subsequent loss of the centriolar microtubules (*Pierron et al., 2023*). In addition, centriole loss in our mutant cells occurs through a stereotyped progression of architectural changes in mitosis, starting with centriole over elongation in prometaphase and culminating with centriole fragmentation and loss. Prolonged mitotic arrest has been reported to result in centriole over elongation through Plk1 activity (*Kong et al., 2020*), and it is possible that a lengthened mitosis, as observed in these mutant cells and cells lacking centrioles (*Farrell et al., 2024*; *Wang et al., 2017*), may also result in over elongation of mutant centrioles with just A-tubules. In addition, we note that CPAP has an expanded domain in mutant centrioles compared to controls (*Figure 4*, *Vásquez-Limeta et al., 2022*). CPAP is involved in slow processive microtubule growth (*Sharma et al., 2016*) and its loss results in centriole fragmentation (*Vásquez-Limeta et al., 2022*), and it is possible that CPAP mislocalization may also contribute to over elongation of these mutant centrioles. Future work will determine the molecular mechanisms by which mutant centrioles lacking triplet microtubules are disassembled through the cell cycle.

Finally, we note that mutant human centrioles lacking compound microtubules bear similarities to the centrioles of *Drosophila* and *C. elegans* embryos, which have evolved to lack triplet microtubules and have cartwheels extending the entire length of the centriole (*González et al., 1998*; *Pelletier et al., 2006*; *Woglar et al., 2022*). Embryonic centrioles in both species are shorter than that of other organisms, and helical inner scaffolds have not been reported. In both species, these diminished centrioles participate in mitosis, can duplicate their centrioles, and serve as basal bodies for sensory cilia. We speculate that centrioles with triplet microtubules and the proteins they anchor, including the inner scaffold, may be required for centriole function in organisms with motile cilia, perhaps to help stabilize the basal body against ciliary movement. Such activity has been described for *Tetrahymena* basal bodies, and mutating an inner scaffold protein, Poc1, results in abnormal bending within basal bodies (*Junker et al., 2022*). Further supporting this hypothesis, *Drosophila* spermatocytes, one of the few cells within this species with motile cilia, have basal bodies with triplet microtubules (*González et al., 1998*). We note that these spermatocytes likely form triplet microtubules in an alternative manner, as *Drosophila* lacks delta-tubulin or epsilon-tubulin.

In conclusion, this work, along with our previously published study, identifies proteins required for the formation or maintenance of the centriolar triplet microtubules and maps the requirements of these proteins and the triplets in centriole architecture. Together, these results pave the way for deeper molecular understanding of the mechanisms by which the triplet microtubules are formed and maintained reproducibly within cells to form robust centrioles and cilia.

# Materials and methods

**Key resources table**

| Reagent type (species) or resource | Designation | Source or reference | Identifiers | Additional information |
|---|---|---|---|---|
| Chemical compound, drug | Sodium acrylate | AK Scientific or Sigma Aldrich | AK Sci cat# R624, Sigma cat# 408220 | There is batch to batch variability in acrylate purity |
| Chemical compound, drug | 40% Acrylamide | Sigma Aldrich | Cat# A4058 | |
| Chemical compound, drug | 36–38% Formaldehyde | Sigma Aldrich | Cat# F8775 | |
| Chemical compound, drug | N,N'-Methylenebisacrylamide solution (BIS) | Sigma Aldrich | Cat# M1533 | |
| Chemical compound, drug | Ammonium Persulfate (APS) | Bio-Rad | Cat# 1610700 | |
| Chemical compound, drug | N,N,N',N'-Tetramethylethylenediamine (TEMED) | Bio-Rad | Cat# 1610800 | |
| Chemical compound, drug | Sodium dodecyl sulfate (SDS) | Sigma Aldrich | Cat# 75746 | |
| chemical compound, drug | Sodium chloride (NaCl) | Sigma Aldrich | Cat# S9888 | |
| Chemical compound, drug | Tris base | Sigma Aldrich | Cat# 93362 | |
| Chemical compound, drug | Potassium chloride (KCl) | Sigma Aldrich | Cat# P3911 | |
| Chemical compound, drug | Triton X-100 | Sigma Aldrich | Cat# T8787 | |
| Chemical compound, drug | DL-Dithiothreitol (DTT) | Sigma Aldrich | Cat# D9779 | |
| Chemical compound, drug | Glycine | Thermo Fisher Scientific | Cat# BP381-5 | |
| Chemical compound, drug | Tween 20 | Sigma Aldrich | Cat# P1379 | |
| Antibody | Anti-acetlyated- tubulin, clone 6-11B-1 (monoclonal mouse IgG2b) | Sigma Aldrich | Cat# T6793, RRID:AB_477585 | UExM (1:1000) |
| Antibody | Anti-acetyl-alpha-tubulin, Lys40 (rabbit polyclonal) | Cell Signaling Technology | Cat# 5335, RRID:AB_10544694 | UExM 1:100 |
| Antibody | Anti-centrin, clone 20H5 (monoclonal mouse IgG2a) | EMD Millipore | RRID:AB_10563501 | IF 1:200, UExM 1:500 |
| Antibody | Anti-centrin3, clone 3e6 (monoclonal mouse IgG2b) | Novus Biological | RRID:AB_537701 | UExM 1:1000 |
| Antibody | Anti-CENPJ (rabbit polyclonal) | Proteintech | Cat# 11517–1-AP, RRID:AB_2244605 | WB 1:1000 UExM 1:500 |
| Antibody | Anti-Cep44 (rabbit polyclonal) | Proteintech | Cat# 24457–1-AP, RRID:AB_2879557 | UExM 1:100 |
| Antibody | Anti-Cep120 (rat polyclonal) | Gift from Moe Mahjoub | | *Betleja et al., 2018* |
| Antibody | Anti-Cep135 (rabbit polyclonal) | Proteintech | Cat# 24428–1-AP, RRID:AB_2879543 | UExM 1:500 |
| Antibody | Anti-Cep295 (rabbit polyclonal) | Sigma-Aldrich | Cat# HPA038596, RRID:AB_10672720 | UExM 1:1000 |
| Antibody | Anti-CP110 (rabbit polyclonal) | Proteintech | Cat# 12780–1-AP, RRID:AB_10638480 | IF 1:200, UExM 1:2000 |

*Continued on next page*

*Continued*

| Reagent type (species) or resource | Designation | Source or reference | Identifiers | Additional information |
|---|---|---|---|---|
| Antibody | Anti-Flag, clone M2 (monoclonal mouse IgG1) | Sigma-Aldrich | Cat# F1804, RRID:AB_262044 | WB 1:2000, UExM 1:500 |
| Antibody | Anti-gamma-tubulin, clone GTU-88 (monoclonal mouse IgG1) | Sigma-Aldrich | RRID:AB_477584 | IF 1:1000, UExM 1:500 |
| Antibody | Anti-PCNA (monoclonal mouse IgG2a) | BioLegend | RRID:AB_314692 | UExM 1:500 |
| Antibody | Anti-POC5 (rabbit polyclonal) | Bethyl Laboratories | RRID:AB_10949152 | IF 1:500 |
| Antibody | Anti-POC5 (rabbit polyclonal) | Thermo Fisher Scientific | Cat# A303-341A, RRID:AB_10971172 | WB 1:1000 UExM 1:500 |
| Antibody | Anti-polyglutamylation, clone GT335 (monoclonal mouse IgG1) | AdipoGen | Cat# AG-20B-0020, RRID:AB_2490210 | UExM 1:500 |
| Antibody | Anti-polyglutamylate-chain, polyE (rabbit polyclonal) | AdipoGen | Cat# AG-25B-0030, RRID:AB_2490540 | UExM 1:500 |
| Antibody | Anti-SASS6 (monoclonal mouse IgG2b) | Santa Cruz | Cat# sc-81431, RRID:AB_1128357 | IF, WB, UExM 1:200 |
| Antibody | Anti-STIL (rabbit polyclonal) | Abcam | Cat# ab89314, RRID:AB_2197878 | WB 1:2000 UExM 1:500 |
| Antibody | Anti-V5 (monoclonal mouse IgG2a) | Thermo Fisher Scientific | Cat# R960-25, RRID:AB_2556564 | WB 1:1000 UExM 1:100 |
| Antibody | Anti-WDR90 (rabbit polyclonal) | Thermo Fisher Scientfic | Cat# PA5-61943, RRID:AB_2649628 | UExM 1:100 |
| Antibody | Goat anti-mouse IgG1, 488 | Thermo Fisher Scientific | Cat# A21121, RRID:AB_2535764 | UExM 1:1000 |
| Antibody | Goat anti-Mouse IgG2a, 488 | Thermo Fisher Scientific | Cat# A-21131, RRID:AB_2535771 | UExM 1:1000 |
| Antibody | Goat anti-Mouse IgG2b, 488 | Thermo Fisher Scientific | Cat# A-21141, RRID:AB_2535778 | UExM 1:1000 |
| Antibody | Goat anti-rabbit IgG (H+L), 488 | Thermo Fisher Scientific | Cat# A-11034, RRID:AB_2576217 | UExM 1:1000 |
| Antibody | Goat anti-Mouse IgG1, 568 | Thermo Fisher Scientific | Cat# A-21124, RRID:AB_2535766 | UExM 1:500 |
| Antibody | Goat anti-Mouse IgG2a, 568 | Thermo Fisher Scientific | Cat# A-21134, RRID:AB_2535773 | UExM 1:500 |
| Antibody | Goat anti-Mouse IgG2b, 568 | Thermo Fisher Scientific | Cat# A-21144, RRID:AB_2535780 | UExM 1:500 |
| antibody | Goat anti-rabbit IgG (H+L), 568 | Thermo Fisher Scientific | Cat# A-11036, RRID:AB_10563566 | UExM 1:500 |
| Antibody | Goat anti-Mouse IgG3, 594 | Thermo Fisher Scientific | Cat# A-21155, RRID:AB_2535785 | UExM 1:500 |
| Antibody | Goat anti-rat IgG (H+L), 594 | Thermo Fisher Scientific | Cat# A-11007, RRID:AB_10561522 | UExM 1:500 |
| Antibody | Goat anti-Mouse IgG1, 647 | Thermo Fisher Scientific | Cat# A-21240, RRID:AB_2535809 | UExM 1:500 |
| Antibody | Goat anti-Mouse IgG2a, 647 | Thermo Fisher Scientific | Cat# A-21241, RRID:AB_2535810 | UExM 1:500 |
| antibody | Goat anti-Mouse IgG2b, 647 | Thermo Fisher Scientific | Cat# A-21242, RRID:AB_2535811 | UExM 1:500 |
| Antibody | Goat anti-rabbit IgG (H+L), 647 | Thermo Fisher Scientific | Cat# A32733, RRID:AB_2633282 | UExM 1:500 |

*Continued on next page*

*Continued*

| Reagent type (species) or resource | Designation | Source or reference | Identifiers | Additional information |
|---|---|---|---|---|
| Antibody | Rabbit Anti-TUBD1 (rabbit polyclonal) | Sigma-Aldrich | Cat# HPA027090, RRID:AB_1858457 | WB 1:1000 |
| Antibody | Rabbit Anti-TUBE1 (rabbit polyclonal) | Sigma-Aldrich | Cat# HPA032074, RRID:AB_10601216 | WB 1:1000 |
| Antibody | rabbit anti C14orf80 (rabbit polyclonal) | Sigma-Aldrich | Cat# HPA039049, RRID:AB_2676320 | WB 1:1000 |
| Antibody | rabbit anti C16orf59 (rabbit polyclonal) | Sigma-Aldrich | Cat# HPA055389, RRID:AB_2732595 | WB 1:1000 |
| Antibody | 680 Donkey anti rabbit (H+L) | Thermo Fisher Scientific | Cat# A10043, RRID:AB_2534018 | WB 1:20,000 |
| Antibody | 800 Donkey anti rabbit (H+L) | Li-COR | Cat# 926–32213, RRID:AB_621848 | WB 1:20,000 |
| Antibody | 680 Donkey anti mouse (H+L) | Thermo Fisher Scientific | Cat# A10038, RRID:AB_11180593 | WB 1:20,000 |
| Antibody | 800 Donkey anti mouse (H+L) | Li-COR | Cat# 926–32212, RRID:AB_621847 | WB 1:20,000 |

## Cell lines and cell culture

Human hTERT RPE-1 *TP53*$^{-/-}$ cells were a gift from Meng-Fu Bryan Tsou (Memorial Sloan Kettering Cancer Center) and were cultured in DMEM/F-12 (Corning) supplemented with 10% Cosmic Calf Serum (CCS; HyClone). Human HEK293T cells for lentivirus production (see below) were obtained from the ATCC and cultured in DMEM (Corning) supplemented with 10% CCS. hTERT RPE-1 and HEK293T/17 cells were authenticated using STR profiling using CODIS loci. All other cell lines used were derived from hTERT RPE-1 *TP53*$^{-/-}$ cells. Stable *TP53*$^{-/-}$; *TEDC1*$^{-/-}$ and *TP53*$^{-/-}$; *TEDC2*$^{-/-}$ knockout cell lines were made in the hTERT RPE-1 *TP53*$^{-/-}$ cells by CRISPR/Cas9 (see below). For rescue experiments, clonal knockout cell lines were rescued using lentiviral transduction (see below). All cells were cultured at 37 °C under 5% $CO_2$, and are mycoplasma-free (*Uphoff and Drexler, 2011*).

## Generation of *TEDC1*$^{-/-}$ and *TEDC2*$^{-/-}$ cells and rescue cell lines

*TEDC1*$^{-/-}$ and *TEDC2*$^{-/-}$ cells were generated by CRISPR/Cas9 mediated gene editing using a recombinantly produced, purified Cas9 protein (Cas9-NLS, QB3 Macrolab, Berkeley) and chemically synthetized two-component gRNA (crRNA:tracrRNA, Alt-R CRISPR-Cas9 system, IDT). For increased efficiency, two gRNAs, both targeting the 5' end of each gene, were used at the same time. Target sequences were: 5'-CGCCAAGTTCGACCGTCCGG-3' and 5'-CGTCCAATCACCGCACGGGC-3' for TEDC1, and 5'-CGCACAGCGACAATTGCAAT-3' and 5'-CACCGGCGCGAGCAGCCCGC-3' for TEDC2.

Lyophilized RNA oligos were reconstituted according to the instructions provided by the manufacturer (IDT). Briefly, oligos were reconstituted in the duplex buffer at a concentration of 200 µM. To anneal crRNA with tracrRNA, 3 µl of each (600 pmol) were mixed, heated to 95 °C, and transferred to room temperature to gradually cool. Pre-complexed crRNA and tracrRNA (550 pmol) were mixed with purified Cas9 (360 pmol), diluted with PBS to a total volume of 25 µl and incubated for 15 min at room temperature to form ribonucloprotein complexes (RNPs).

RPE1 *TP53*$^{-/-}$ cells stably expressing GFP-centrin (*Wang et al., 2017*) were electroporated in a home-made electroporation buffer (*Zhang et al., 2014*) using Amaxa Nucleofector II (Lonza). Cells were electroporated with an equal mix of two RNPs: 50 µl of RNPs mixture was added to 2x10$^6$ cells in 200 µl electroporation buffer. To facilitate the identification of electroporated cells, an mRuby2 expressing plasmid (pcDNA3-mRuby2, plasmid pTS3994) was electroporated together with RNPs.

Two days after electroporation, cells expressing mRuby2 were sorted using FACS, and single cells were plated into 96-well plates in conditioned media. Surviving clones were genotyped by PCR of genomic DNA and screened for phenotype based on centrin-GFP expression.

Primers used for genotyping were: 5'CCCTGCCGACGCAGTGATTGG3' and 5'CAGGGAGTGGCG AGAGCACAC3' for TEDC1 and 5'CTTGCCCGCAAGGAGGGAGAGA3' and 5'GCAGGGCCCAGC CCAAACAGA3' for TEDC2.

To rescue the mutations, Halotag-3xFlag-tagged TEDC1 or APEX-V5-tagged TEDC2 were introduced into the mutant cells using lentiviral transduction as described below.

## Lentivirus production and viral transduction

Recombinant lentiviruses were made by cotransfection of HEK293T cells with the respective transfer vectors (TEDC1-Halotag-3xFlag and TEDC2-V5-APEX2), second-generation lentiviral cassettes (packaging vector psPAX2, pTS3312 and envelope vector pMD2.G, pTS3313) using calcium phosphate-mediated transfection. Briefly, transfection mixture was made with CaCl2, 2 x HBS (50 mM Hepes, 10 mM KCl, 12 mM dextrose, 280 mM NaCl, 1.5 mM Na2HPO4x7H2O, pH 7.05), and plasmids. Cells were treated with 25 µM chloroquine immediately before transfection, then the transfection mixture was added to cells. The medium was changed 5–6 hr after transfection, and viral supernatant was harvested after an additional 48 and 72 hr. Recipient cells (RPE-1 $TP53^{-/-}$; $TEDC1^{-/-}$ and $TP53^{-/-}$; $TEDC2^{-/-}$ and $TP53^{-/-}$; $TUBD1^{-/-}$ and $TP53^{-/-}$; $TUBE1^{-/-}$) were transduced with viral supernatant and 8 µg/mL Sequabrene. Transduced cells were expanded to 10 cm dishes.

## Immunofluorescence

Cells were grown on poly-L-lysine-coated #1.5 glass coverslips (Electron Microscopy Sciences). Cells were fixed with −20 °C methanol for 15 min. Coverslips were then washed with PBS for 10 min and blocked with PBS-BT (3% BSA, 0.1% Triton X-100, 0.02% sodium azide in PBS) for 30 min. Coverslips were incubated with primary antibodies diluted in PBS-BT for 1 hr, washed with PBS-BT, incubated with secondary antibodies and DAPI diluted in PBS-BT for 1 hr, then washed again. Samples were mounted using Mowiol (Polysciences) in glycerol containing 1,4,-diazobicycli-[2.2.2]octane (DABCO, Sigma-Aldrich) antifade.

## Cell cycle synchronization

For cell cycle analyses in *Figure 1*, cells were seeded onto coverslips, then synchronized in G0/G1 by serum withdrawal for 24 hr, or in G2 with 10 µM RO-3306 (Adipogen) for 24 hr. Cells were fixed for immunofluorescence and analyzed for centrin/CP110 presence. Three biological replicates were performed.

For *Figures 4 and 5*, mitotic shakeoff was performed on asynchronously growing cells. One pre-shake was performed to improve synchronization. Cells were fixed for U-ExM and expanded as below.

## Expansion microscopy

### Ultrastructure expansion microscopy (U-ExM)

Cells were grown on poly-D-lysine-coated #1.5 glass coverslips (Electron Microscopy Sciences) and fixed with −20 °C methanol for 15 min, then washed with PBS. U-ExM was performed as previously described (*Gambarotto et al., 2019*): coverslips were incubated overnight in an acrylamide–formaldehyde anchoring solution (AA/FA; 0.7% formaldehyde, 1% acrylamide in PBS) at 37 °C. Gelation was allowed to proceed in monomer solution (19% sodium acrylate, 10% acrylamide, 0.1% bis-acrylamide, 0.5% ammonium persulfate-APS, 0.5% TEMED) for 1 hr at 37 °C. Gels were heated in denaturation buffer (200 mM SDS, 200 mM NaCl, 50 mM Tris-HCl pH 9) at 95 °C for 1 hr. After denaturation buffer was removed, gels were washed with multiple water rinses and allowed to expand in water at room temperature overnight. Small circles of each expanded gel (~5 mm in diameter) were excised and incubated with primary antibodies diluted in PBS-BT (3% BSA, 0.1% Triton X-100 in PBS) on a nutator at 4 °C overnight. The next day, gels were washed three times with PBS-BT buffer and incubated with secondary antibodies and 5 µg/ml DAPI diluted in PBS-BT, protected from light, on a nutator at 4 °C overnight.

For *Figure 4*, *Figure 4—figure supplement 3* and *Figure 4—figure supplement 4* when co-staining with alpha-tubulin, centrioles were fixed with 1.4% formaldehyde and 2% acrylamide for 3–5 hr at 37 °C. U-ExM was performed as described above. Gels were pre-incubated with anti alpha-tubulin antibody at 4 °C overnight prior to staining with other primary antibodies.

### Expansion microscopy as per Kong et al

For *Figure 2—figure supplement 1C,D*, expansion microscopy was performed similar to *Kong et al., 2024*. Coverslips were incubated in 4% formaldehyde in 1 x PBS for 1 hr. The coverslips were then incubated overnight in an acrylamide–formaldehyde anchoring solution (AA/FA; 4% formaldehyde, 30% acrylamide in PBS) at 40 °C. Gelation was allowed to proceed in monomer solution (7% sodium acrylate, 20% acrylamide, 0.04% bis-acrylamide, 0.5% ammonium persulfate-APS, 0.5% TEMED in PBS) for 20 min on ice followed by 1 hr at room temperature. Gels were heated in denaturation buffer (200 mM SDS, 200 mM NaCl, 50 mM Tris-HCl pH 9) at 90 °C for 1 hr. After denaturation buffer was removed, gels were washed with multiple water rinses and allowed to expand in water at room temperature overnight. Small circles of each expanded gel (~5 mm in diameter) were excised and incubated with primary antibodies diluted in PBS-BT (3% BSA, 0.1% Triton X-100 in PBS) at 4 °C overnight. The next day, gels were washed three times with PBS-BT buffer and incubated with secondary antibodies and 5 µg/ml DAPI diluted in PBS-BT, protected from light, at 4 °C overnight.

### Expansion gel imaging (all protocols)

Immunostained gels were washed once with PBS and at least three times with water, and placed in a glass-bottomed 35 mm plate for imaging. All U-ExM images were acquired as *z*-stacks collected at 0.27 µm intervals using a confocal Zeiss Axio Observer microscope (Carl Zeiss) with a PlanApo-Chromat 1.4 NA 63×oil immersion objective, a Yokogawa CSU-W1 (*Figure 2*) or Yokogawa CSU-W1 SoRA head with 2.8 x relay (*Figures 4 and 5*) and a Photometrics Prime BSI express CMOS camera. Slidebook software (Intelligent Imaging Innovations, 3i) was used to control the microscope system. Deconvolution was performed with Microvolution (Cupertino, CA) using a calculated point spread function (PSF) for 10 iterations. ImageJ (FIJI) was used for image analysis (*Schindelin et al., 2012*).

## Centriole measurements

For measuring overall centriole width or length, z-stacks of U-ExM images were measured using ImageJ (FIJI) on maximum projections. Only centrioles that were in perfect longitudinal or cross-section were measured. Three measurements were made per centriole and averaged. Measurements were adjusted for gel expansion factor. Statistical analysis was performed with Graphpad Prism.

For measuring protein position as in *Figure 4—figure supplement 3* and *Figure 4—figure supplement 4*, maximum projections of U-ExM images of longitudinally positioned centrioles were measured using ImageJ (FIJI). The coordinates of the proximal-most and distal-most position for each protein were recorded. Three measurements were made per centriole and averaged. The recorded coordinates were used to calculate the positions of the most proximal and most distal signal for each protein, then graphed from shortest to longest centriole.

Welch's t-test was chosen for statistical analysis in *Figure 4A*, which is an unpaired t test that does not assume that the two datasets have the same variance.

## Transmission electron microscopy

For ultrastructural analysis of centrosomes by TEM, RPE-1 *TP53*[-/-]; *TEDC1*[-/-] and RPE-1 *TP53*[-/-]; *TEDC2*[-/-] cells were synchronized in G2/M with 10 µM RO-3306 for 24 hrs. Cells were trypsinized, resuspended in complete media and centrifuged at 800 × *g* for 5 min. The pellet was collected in a 14 mL tube and fixed in 2% paraformaldehyde/2.5% glutaraldehyde (Ted Pella Inc, Redding, CA) in 100 mM cacodylate buffer, pH 7.2 for 2 hr at room temperature. Samples were washed in cacodylate buffer and postfixed in 1% osmium tetroxide (Ted Pella Inc)/1.5% potassium ferricyanide (Sigma, St. Louis, MO) for 1 hr. Samples were then rinsed extensively in dH$_2$O prior to en bloc staining with 1% aqueous uranyl acetate (Ted Pella Inc) for 1 hr. Following several rinses in dH$_2$O, samples were dehydrated in a graded series of ethanol and embedded in Eponate 12 resin (Ted Pella Inc). Ultrathin sections of 95 nm were cut with a Leica Ultracut UCT ultramicrotome (Leica Microsystems Inc, Bannockburn, IL), stained with uranyl acetate and lead citrate, and viewed on a JEOL 1200 EX transmission electron microscope (JEOL USA Inc, Peabody, MA) equipped with an AMT 8 megapixel digital camera and AMT Image Capture Engine V602 software (Advanced Microscopy Techniques, Woburn, MA).

Symmetrization of TEM images was performed with centrioleJ.

## TEDC1 and TEDC2 pulldowns

Cells stably expressing TEDC1-Halotag-3xFlag or TEDC2-V5-APEX2 were lysed in 50 mM Tris pH7.5, 150 mM NaCl, 1% Triton X-100, 1 mM DTT, Halt protease and phosphatase inhibitor cocktail (ThermoFisher Scientific) for 30 min on ice, then cleared by centrifugation at 21,000 × *g* for 20 min. Protein concentration was determined by Pierce BCA Protein Assay - Reducing Agent Compatible (ThermoFisher Scientific). Each cell lysate was incubated with 25 µL of equilibrated Chromotek Halo-Trap Magnetic Agarose (Proteintech) or Chromotek V5-Trap Magnetic Agarose (Proteintech) for 1 hr at 4 °C on a nutator. Beads were washed using a magnetic separator rack. Elution was performed by adding 80 µL of 2 x SDS loading buffer (100 mM Tris pH 6.8, 4% SDS, 20% glycerol, 100 mM DTT), boiling the beads for 5 min at 95 °C, then separating the eluate with a magnetic separator rack. Samples were loaded on SDS-PAGE and transferred for western blotting. Three biological replicates were performed.

## Western blotting

For *Figure 4—figure supplement 2*, samples were lysed in 50 mM Tris pH7.5, 150 mM NaCl, 1% Triton X-100, 1 mM DTT, Halt protease and phosphatase inhibitor cocktail (Thermo Fisher Scientific) for 30 min on ice, then cleared by centrifugation at 21,000 × *g* for 20 min. Protein concentration was determined by Pierce BCA Protein Assay - Reducing Agent Compatible (Thermo Fisher Scientific). Equal amounts of protein (20–40 µg) were loaded per lane. For *Figure 3*, samples were loaded after pulldowns.

Proteins were separated by SDS-PAGE and transferred to nitrocellulose (LiCOR Biosciences) in transfer buffer (192 mM Glycine, 25 mM Tris, 20% ethanol). Membranes were blocked with 5% milk in TBST (137 mM NaCl, 25 mM Tris, 2.7 mM KCl, 0.1% Tween-20) at room temp for 1 h, then washed three times with TBST for 5 min each wash. Membranes were incubated with primary antibodies overnight at 4°C on a nutator. The next day, membranes were washed three times with TBST for 5 min each wash and incubated with secondary antibodies at room temperature for 2.5 hr. Membranes were washed again with TBST for 5 min each wash and then imaged with the LiCOR Odyssey XF imager and analyzed using Image Studio (LiCOR Biosciences). Three biological replicates were performed.

## Antibodies

Primary antibodies used for immunofluorescence and U-ExM and dilutions in PBS-BT: mouse IgG2b anti-acetylated-tubulin, clone 6-11B-1 (1:1000,Sigma-Aldrich Cat# T6793, RRID:AB_477585), rabbit anti-acetyl-α-tubulin (Lys40) (1:100, Cell Signaling Technology Cat# 5335, RRID:AB_10544694), mouse IgG2b anti-centrin3, clone 3e6 (1:1000, Novus Biological, RRID:AB_537701), mouse IgG2a anti-centrin, clone 20H5 (IF 1:200, UExM 1:500, EMD Millipore, RRID:AB_10563501), rat anti-Cep120 (1:1000, gift from Moe Mahjoub *Betleja et al., 2018*), rabbit anti-Cep135 (1:500, Proteintech Cat# 24428–1-AP, RRID:AB_2879543), rabbit anti-Cep295 (1:1000, Sigma-Aldrich Cat# HPA038596, RRID:AB_10672720), rabbit anti-Cep44 (1:100, Proteintech Cat# 24457–1-AP, RRID:AB_2879557), rabbit anti-CENPJ (1:500, Proteintech Cat# 11517–1-AP, RRID:AB_2244605), rabbit anti-CP110 (IF 1:200, UExM 1:2000, Proteintech Cat# 12780–1-AP, RRID:AB_10638480), mouse IgG1 anti-Flag, clone M2 (1:500, Sigma-Aldrich Cat# F1804, RRID:AB_262044), mouse IgG1 anti-gamma-tubulin, clone GTU-88 (IF 1:1000, UExM 1:500, Sigma-Aldrich, RRID:AB_477584), mouse IgG2a anti-PCNA (1:500, BioLegend, RRID:AB_314692), rabbit anti-POC5 (for IF: 1:500, Bethyl Laboratories, RRID:AB_10949152), rabbit anti-POC5 (for U-ExM: 1:500, Thermo Fisher Scientific Cat# A303-341A (also A303-341A-T), RRID:AB_10971172), mouse IgG1 anti-polyglutamylation, clone GT335 (1:500, AdipoGen Cat# AG-20B-0020, RRID:AB_2490210), rabbit anti-polyglutamate-chain, polyE (1:500, AdipoGen Cat# AG-25B-0030, RRID:AB_2490540), mouse IgG2b anti-SASS6 (1:200, Santa Cruz Cat# sc-81431, RRID:AB_1128357), rabbit anti-STIL (1:500, Abcam Cat# ab89314, RRID:AB_2197878), mouse IgG2a anti-V5 (1:00, Thermo Fisher Scientific Cat# R960-25, RRID:AB_2556564), rabbit anti-WDR90 (1:100, Thermo Fisher Scientific Cat# PA5-61943, RRID:AB_2649628), chicken anti-GFP antibody (Aves Cat# GFP-1020, RRID:AB_10000240).

For immunofluorescence and U-ExM, AlexaFluor conjugated secondary antibodies (ThermoFisher) were diluted 1:1000 in PBS-BT. Goat anti-Mouse IgG1, 488 (1:1000, Thermo Fisher Scientific Cat# A-21121, RRID:AB_2535764), Goat anti-Mouse IgG2a, 488 (1:1000, Thermo Fisher Scientific Cat# A-21131, RRID:AB_2535771), Goat anti-Mouse IgG2b, 488 (1:1000, Thermo Fisher Scientific

Cat# A-21141, RRID:AB_2535778), Goat anti-rabbit IgG (H+L), 488 (1:1000, Thermo Fisher Scientific Cat# A-11034 (also A11034), RRID:AB_2576217), Goat anti-Mouse IgG1, 568 (1:500, Thermo Fisher Scientific Cat# A-21124, RRID:AB_2535766), Goat anti-Mouse IgG2a, 568 (1:500, Thermo Fisher Scientific Cat# A-21134, RRID:AB_2535773), Goat anti-Mouse IgG2b, 568 (1:500, Thermo Fisher Scientific Cat# A-21144, RRID:AB_2535780), Goat anti-rabbit IgG (H+L), 568 (1:500, Thermo Fisher Scientific Cat# A-11036 (also A11036), RRID:AB_10563566), Goat anti-Mouse IgG3, 594 (1:500, Thermo Fisher Scientific Cat# A-21155, RRID:AB_2535785), Goat anti-rat IgG (H+L), 594 (1:500,Thermo Fisher Scientific Cat# A-11007 (also A11007), RRID:AB_10561522), Goat anti-Mouse IgG1, 647 (1:500, Thermo Fisher Scientific Cat# A-21240, RRID:AB_2535809), Goat anti-Mouse IgG2a, 647 (1:500, Thermo Fisher Scientific Cat# A-21241, RRID:AB_2535810), Goat anti-Mouse IgG2b, 647 (1:500, Thermo Fisher Scientific Cat# A-21242, RRID:AB_2535811), Goat anti-rabbit IgG (H+L), 647 (1:500, Thermo Fisher Scientific Cat# A32733, RRID:AB_2633282), Goat anti-Mouse, Star Red (1:200, Abberior Cat# STRED-1001, RRID:AB_3068620), Goat anti-rabbit, Star Orange (1:200, Abberior Cat# STORANGE-1002, RRID:AB_3068622), Goat anti-chicken, Alexa 488 (Thermo Fisher Scientific Cat# A-11039, RRID:AB_2534096).

Primary antibodies used for Western blotting and dilutions in TBST: rabbit anti TUBD1 (1:1000, Sigma-Aldrich Cat# HPA027090, RRID:AB_1858457), rabbit anti TUBE1 (1:1000, Sigma-Aldrich Cat# HPA032074, RRID:AB_10601216), rabbit anti C14orf80 (1:1000, Sigma-Aldrich Cat# HPA039049, RRID:AB_2676320), rabbit anti C16orf59 (1:1000, Sigma-Aldrich Cat# HPA055389, RRID:AB_2732595), mouse IgG2b anti SASS6 (1:200, Santa Cruz Biotech Cat# sc-81431, RRID:AB_1128357), rabbit anti STIL (1:2000, Abcam Cat# ab89314, RRID:AB_2197878), rabbit anti CENPJ/CPAP (1:1000, Proteintech Cat# 11517–1-AP, RRID:AB_2244605), rabbit anti POC5 (1:1000, Thermo Fisher Scientific Cat# A303-341A (also A303-341A-T), RRID:AB_10971172), mouse IgG2a anti V5 (1:1000, Thermo Fisher Scientific Cat# R960-25, RRID:AB_2556564), mouse IgG1 anti Flag, clone M2 (1:2000, Sigma-Aldrich Cat# F1804, RRID:AB_262044). Secondary antibodies used for Western blotting: 680 Donkey anti rabbit (H+L) (1:20,000, Thermo Fisher Scientific Cat# A10043, RRID:AB_2534018), 800 Donkey anti rabbit (H+L) (1:20,000, Li-COR Cat# 926–32213, RRID:AB_621848), 680 Donkey anti mouse (H+L) (1:20,000, Thermo Fisher Scientific Cat# A10038, RRID:AB_11180593), 800 Donkey anti mouse (H+L) (1:20,000, Li-COR Cat# 926–32212, RRID:AB_621847).

## Novel materials availability statement

The cell lines generated in this work are available through contacting the corresponding author (Jennifer T. Wang, Department of Biology, Washington University in St. Louis).

## Acknowledgements

This work was supported by R00GM131024 to JTW., R35GM130286 to TS and Washington University in St. Louis startup funds (to JTW). We thank Wandy Beatty of the Washington University Molecular Microbiology Imaging Facility for assistance with transmission electron microscopy, Moe Mahjoub (Washington University School of Medicine) for the gift of the CEP120 and goat anti-rat antibodies, and Meng-Fu Bryan Tsou (Memorial Sloan Kettering Cancer Center) for the gifts of RPE-1 *TP53*[-/-] and RPE-1 *TP53*[-/-]; *SASS6*[-/-] cells. We thank the Stanford cytoskeleton group, WashU centrosome/cilia group, members of the Stearns lab, and David Breslow for helpful discussions. We also thank Hani Zaher's and Joe Jez's labs for hosting the Wang lab during renovations and Larry Galloway for help with Python.

## Additional information

### Funding

| Funder | Grant reference number | Author |
| --- | --- | --- |
| National Institutes of Health | R00GM131024 | Jennifer T Wang |

| Funder | Grant reference number | Author |
|---|---|---|
| National Institutes of Health | R35GM130286 | Tim Stearns |

The funders had no role in study design, data collection and interpretation, or the decision to submit the work for publication.

## Author contributions

Rachel Pudlowski, Conceptualization, Data curation, Formal analysis, Investigation, Visualization, Methodology, Writing – original draft, Writing – review and editing; Lingyi Xu, Investigation, Visualization, Methodology, Writing – original draft, Writing – review and editing; Ljiljana Milenkovic, Investigation, Methodology, Writing – original draft, Writing – review and editing; Chandan Kumar, Investigation, Methodology, Writing – review and editing; Katherine Hemsworth, Data curation, Investigation, Writing – review and editing; Zayd Aqrabawi, Data curation, Formal analysis, Writing – review and editing; Tim Stearns, Funding acquisition, Writing – original draft, Writing – review and editing; Jennifer T Wang, Conceptualization, Resources, Data curation, Formal analysis, Supervision, Funding acquisition, Validation, Investigation, Visualization, Methodology, Writing – original draft, Project administration, Writing – review and editing

## Author ORCIDs

Rachel Pudlowski  https://orcid.org/0009-0002-7767-1147
Tim Stearns  https://orcid.org/0000-0002-0671-6582
Jennifer T Wang  https://orcid.org/0000-0002-8506-5182

Reviewer #1 (Public review): https://doi.org/10.7554/eLife.98704.3.sa1
Reviewer #2 (Public review): https://doi.org/10.7554/eLife.98704.3.sa2
Reviewer #3 (Public review): https://doi.org/10.7554/eLife.98704.3.sa3
Author response https://doi.org/10.7554/eLife.98704.3.sa4

# Additional files

## Supplementary files

MDAR checklist

## Data availability

All data generated or analyzed in this work are included in the manuscript and the supporting source data files.

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
