## [Editor Report · eLife Assessment]

The study by Pudlowski et al. shows that a previously-identified protein complex, composed of delta- and epsilon-tubulin together with TEDC1 and TEDC2, functions in generating centriolar triplet microtubules, and that this is crucial for the proper formation of centriolar subdomains and the stability of centrioles throughout the cell cycle. This is an **important** study that advances our understanding of centriole biogenesis and structure and is supported by **convincing** evidence based on knockout cell lines, immunoprecipitation, and ultrastructure expansion microscopy. The work is of interest to cell biologists, in particular researchers with interest in centrosome biology.

---

## [Referee Report · Reviewer #1 (Public review)]

Summary:

The study by Pudlowski et al. investigates how the intricate structure of centrioles is formed by studying the role of a complex formed by delta- and epsilon-tubulin and the TEDC1 and TEDC2 proteins. For this they employ knockout cell lines, EM and ultrastructure expansion microscopy as well as pull-downs. Previous work has indicated a role of delta- and epsilon-tubulin in triplet microtubule formation. Without triplet microtubules centriolar cylinders can still form, but are unstable, resulting is futile rounds of de novo centriole assembly during S phase and disassembly during mitosis. Here the authors show that all four proteins function as a complex and knockout of any of the four proteins results in the same phenotype. They further find that mutant centrioles lack inner scaffold proteins and contain an extended proximal end including markers such as SAS6 and CEP135, suggesting that triplet microtubule formation is linked to limiting proximal end extension and formation of the central region that contains the inner scaffold. Finally, they show that mutant centrioles seem to undergo elongation during early mitosis before disassembly, although it is not clear if this may also be due to prolonged mitotic duration in mutants.

Strengths:

Overall this is a well-performed study, well presented, with conclusions supported by convincing data based on knockout cell lines, rescue experiments, and detailed quantifications.

Weaknesses:

Most weaknesses have been addressed in the revised version. The precise mapping of TED complex proteins to centrioles remains challenging with the available tools but has been addressed through the use of several complementary super-resolution techniques.

---

## [Referee Report · Reviewer #2 (Public review)]

Summary:

In this article, the authors study the function of TEDC1 and TEDC2, two proteins previously reported to interact with TUBD1 and TUBE1. Previous work by the same group had shown that TUBD1 and TUBE1 are required for centriole assembly and that human cells lacking these proteins form abnormal centrioles that only have singlet microtubules that disintegrate in mitosis. In this new work, the authors demonstrate that TEDC1 and TEDC2 depletion results in the same phenotype with abnormal centrioles that also disintegrate into mitosis. In addition, they were able to localize these proteins to the proximal end of the centriole, a result not previously achieved with TUBD1 and TUBE1, providing a better understanding of where and when the complex is involved in centriole growth.

Strengths:

The results are very convincing, particularly the phenotype, which is the same as previously observed for TUBD1 and TUBE1. The U-ExM localization is also convincing: despite a signal that's not very homogeneous, it's clear that the complex is in the proximal region of the centriole and procentriole. The phenotype observed in U-ExM on the elongation of the cartwheel is also spectacular and opens the question of the regulation of the size of this structure. The authors also report convincing results on direct interactions between TUBD1, TUBE1, TEDC1, and TEDC2, and an intriguing structural prediction suggesting that TEDC1 and TEDC2 form a heterodimer that interacts with the TUBD1- TUBE1 heterodimer.

Comments on revisions:

I would like to thank the authors for their work and for thoroughly addressing most of my questions. I extend my congratulations to the authors for this excellent and impactful article.

---

## [Referee Report · Reviewer #3 (Public review)]

Summary:

Human cells deficient in delta-tubulin or epsilon-tubulin form unstable centrioles, which lack triplet microtubules and undergo a futile formation and disintegration cycle. In this study, the authors show that human cells lacking the associated proteins TEDC1 or TEDC2 have these identical phenotypes. They use genetics to knockout TEDC1 or TEDC2 in p53-negative RPE-1 cells and expansion microscopy to structurally characterize mutant centrioles. Biochemical methods and AlphaFold-multimer prediction software are used to investigate interactions between tubulins and TEDC1 and TEDC2.

The study shows that mutant centrioles are built only of A tubules, which elongate and extend their proximal region, fail to incorporate structural components, and finally disintegrate in mitosis. In addition, they demonstrate that delta-tubulin or epsilon-tubulin and TEDC1 and TEDC2 form one complex and that TEDC1 TEDC2 can interact independently of tubulins. Finally, they show that localization of four proteins is mutually dependent.

Strengths:

The results presented here are convincing, exciting, and important, and the manuscript is well-written. The study shows that delta-tubulin, epsilon-tubulin, TEDC1, and TEDC2 function together to build a stable and functional centriole, significantly contributing to the field and our understanding of the centriole assembly process.

Weaknesses:

The ultrastructural characterization of TEDC1 and TEDC2 in centrosomes remains challenging. Nevertheless, it is evident that these proteins occupy growing centrioles and the proximal parts of mother centrioles.

Comments on revisions:

The authors have done a great job extending the original experiments and measurements and answering outstanding questions.

---

## [Author Response]

The following is the authors’ response to the original reviews.

**Public Reviews:**

**Reviewer #1 (Public Review):**
Summary:The study by Pudlowski et al. investigates how the intricate structure of centrioles is formed by studying the role of a complex formed by delta- and epsilon-tubulin and the TEDC1 and TEDC2 proteins. For this, they employ knockout cell lines, EM, and ultrastructure expansion microscopy as well as pull-downs. Previous work has indicated a role of delta- and epsilon-tubulin in triplet microtubule formation. Without triplet microtubules centriolar cylinders can still form, but are unstable, resulting in futile rounds of de novo centriole assembly during S phase and disassembly during mitosis. Here the authors show that all four proteins function as a complex and knockout of any of the four proteins results in the same phenotype. They further find that mutant centrioles lack inner scaffold proteins and contain an extended proximal end including markers such as SAS6 and CEP135, suggesting that triplet microtubule formation is linked to limiting proximal end extension and formation of the central region that contains the inner scaffold. Finally, they show that mutant centrioles seem to undergo elongation during early mitosis before disassembly, although it is not clear if this may also be due to prolonged mitotic duration in mutants.Strengths:Overall this is a well-performed study, well presented, with conclusions mostly supported by the data. The use of knockout cell lines and rescue experiments is convincing.Weaknesses:In some cases, additional controls and quantification would be needed, in particular regarding cell cycle and centriole elongation stages, to make the data and conclusions more robust.

We thank the reviewer for these comments and have improved our analyses of these as detailed below.

**Reviewer #2 (Public Review):**
Summary:In this article, the authors study the function of TEDC1 and TEDC2, two proteins previously reported to interact with TUBD1 and TUBE1. Previous work by the same group had shown that TUBD1 and TUBE1 are required for centriole assembly and that human cells lacking these proteins form abnormal centrioles that only have singlet microtubules that disintegrate in mitosis. In this new work, the authors demonstrate that TEDC1 and TEDC2 depletion results in the same phenotype with abnormal centrioles that also disintegrate into mitosis. In addition, they were able to localize these proteins to the proximal end of the centriole, a result not previously achieved with TUBD1 and TUBE1, providing a better understanding of where and when the complex is involved in centriole growth.Strengths:The results are very convincing, particularly the phenotype, which is the same as previously observed for TUBD1 and TUBE1. The U-ExM localization is also convincing:despite a signal that's not very homogeneous, it's clear that the complex is in the proximal region of the centriole and procentriole. The phenotype observed in U-ExM on the elongation of the cartwheel is also spectacular and opens the question of the regulation of the size of this structure. The authors also report convincing results on direct interactions between TUBD1, TUBE1, TEDC1, and TEDC2, and an intriguing structural prediction suggesting that TEDC1 and TEDC2 form a heterodimer that interacts with the TUBD1- TUBE1 heterodimer.Weaknesses:The phenotypes observed in U-ExM on cartwheel elongation merit further quantification, enabling the field to appreciate better what is happening at the level of this structure.

We thank the reviewer for these comments and have improved our analyses of cartwheel elongation as detailed below.

**Reviewer #3 (Public Review):**
Summary:Human cells deficient in delta-tubulin or epsilon-tubulin form unstable centrioles, which lack triplet microtubules and undergo a futile formation and disintegration cycle. In this study, the authors show that human cells lacking the associated proteins TEDC1 or TEDC2 have these identical phenotypes. They use genetics to knockout TEDC1 or TEDC2 in p53negative RPE-1 cells and expansion microscopy to structurally characterize mutant centrioles. Biochemical methods and AlphaFold-multimer prediction software are used to investigate interactions between tubulins and TEDC1 and TEDC2.The study shows that mutant centrioles are built only of A tubules, which elongate and extend their proximal region, fail to incorporate structural components, and finally disintegrate in mitosis. In addition, they demonstrate that delta-tubulin or epsilon-tubulin and TEDC1 and TEDC2 form one complex and that TEDC1 TEDC2 can interact independently of tubulins. Finally, they show that the localization of four proteins is mutually dependent.Strengths:The results presented here are mostly convincing, the study is exciting and important, and the manuscript is well-written. The study shows that delta-tubulin, epsilon-tubulin, TEDC1, and TEDC2 function together to build a stable and functional centriole, significantly contributing to the field and our understanding of the centriole assembly process.Weaknesses:The ultrastructural characterization of TEDC1 and TEDC2 obtained by U-ExM is inconclusive. Improving the quality of the signals is paramount for this manuscript.

We thank the reviewer for these comments and have improved our imaging of TEDC1 and TEDC2 localization, as detailed below.

**Recommendations for the authors:**

**Reviewing Editor (Recommendations For The Authors):**
The reviewers agreed that the conclusions are largely supported by solid evidence, but felt that improving the following aspects would make some of the data more convincing:(1) The UExM localizations of TEDC1/2 are not very convincing and the reviewers suggest to complement these with alternative super-resolution approaches (e.g. SIM) and/or different labeling techniques such as pre-expansion labeling using STAR red/orange secondaries (also robust for SIM and STED), use of the Halo tag, different tag antibodies, etc

We thank the reviewers for these recommendations and have adapted two of these strategies to improve our imaging of TEDC1 and TEDC2 localization. First, we used an alternative super-resolution approach, a Yokogawa CSU-W1 SoRA confocal scanner (resolution = 120 nm) and imaged cells grown on coverslips (not expanded). We found that TEDC1 and TEDC2 localize to procentrioles and the proximal end of parental centrioles (Fig 2 – Supplementary Figure 1a, b). Second, we used a recently described expansion gel chemistry (Kong et al., Methods Mol Biol 2024) combined with Abberior Star red and orange secondary antibodies. This technique resulted in robust signal at centrosomes and in the cytoplasm and indicated that TEDC1 and TEDC2 localize near the centriole walls of procentrioles and the proximal region of parental centrioles, near CEP44 (Fig 2 – Supplementary Figure 1c, d). These results complement and support our initial observations (Fig 2C, D) and we have edited the text to reflect this (lines 157-163). We also note that these Flag tag and V5 tag primary antibodies are specific and have little background signal in all applications (Fig 2 – Supplementary Fig 1E-J), while other commercially available antibodies against these tags did exhibit non-specific signal.

(2) The cell cycle classifications of centrioles would strongly benefit, apart from a better description, from adding quantifications of average centriole length at a given stage based on tubulin staining (not acTub).

We thank the reviewers for these recommendations. We have added an improved description of our cell cycle analyses (lines 234-237). We have also added new analyses for centriole length as measured by staining with alpha-tubulin (Fig 4 – Supp 3 and Fig 4 – Supp 4). We find that in all mutants, acetylated tubulin elongates along with alpha-tubulin in a similar way as control centrioles.

**Reviewer #1 (Recommendations For The Authors):**
Specific points:(1) The introduction is a bit oddly structured. About halfway through it summarizes what is going to be presented in the study, giving the impression that it is about to conclude, but then continues with additional, detailed introduction paragraphs. Overall, the authors may also want to consider making it more concise.

We thank the reviewer for these suggestions and have shortened and restructured the introduction for clarity and conciseness.

(2) The text should explain to the non-expert reader why endogenous proteins are not detected and why exogenously expressed, tagged versions are used. Related to this, the authors state overexpression, but what is this assessment based on? Does expression at the endogenous level also rescue? At least by western blot, these questions should be addressed.

In the text, we have added clarification about why endogenous proteins were not detected for immunofluorescence (lines 149-151). To quantify the overexpression, we have added Western blots of TEDC1 and TEDC2 to Fig 1 – Supplementary Figure 1E,F. We note that endogenous levels of both proteins are very low, and the rescue constructs are overexpressed 20 to 70 fold above endogenous levels.

(3) The figures should clearly indicate when tagged proteins are used and detected.Currently, this info is only found in the legends but should be in the figure panels as well.

We have made these changes to the figure panels in Fig 2, Fig 2 – Supp 1, and Fig 3.

(4) I could not find a description and reference to Figure 2 Supplement 2 and 3.

We have replaced these supplements with new supplementary figures for TEDC1 and TEDC2 localization (Fig 2 – Supp 1).

(5) The multiple bands including unspecific (?) bands should be labeled to guide the reader in the western blots.

We have labeled nonspecific bands in our Western blots with asterisks (Fig 1 – Supp 1, Fig 3)

(6) The alphafold prediction suggests that TUBD1 can bind to the TED complex in the absence of TUBE1 can this be shown? This would be a nice validation of the predicted architecture of the complex. I also missed a bit of a discussion of the predicted architecture. How could it be linked to triplet microtubule formation? Is the latest alphafold version 3 adding anything to this analysis?

In our pulldown experiments, we found that TUBD1 cannot bind to TEDC1 or TEDC2 in the absence of TUBE1 (Fig 3C, D, IB: TUBD1). We performed this experiment with three biological replicates and found the same result. It is possible that TUBD1 and TUBE1 form an intact heterodimer, similar to alpha-tubulin and beta-tubulin, and this will be an exciting area of future research.

We have added new analysis from AlphaFold3 (Fig 3 – Supp 1B). AlphaFold3 predicts a similar structure as AlphaFold Multimer.

We have also added additional discussion about the AlphaFold prediction to the text (lines 220-222, 365-367). Thanks to the reviewer for pointing out this oversight.

(7) I suggest briefly explaining in the text how cells and centrioles at different cell cycle stages were identified. I found some info in the legend of Figure 1, but no info for other figures or in the text. Related to this, how are procentrioles defined in de novo formation? There is no parental centriole to serve as a reference.

We have added a brief explanation of the synchronization and identification in lines 234-237. We have also clarified the text regarding *de novo* centrioles, and now term these “de novo centrioles in the first cell cycle after their formation” (lines 271-272).

(8) Related to point 7: using acetylated tubulin as a universal length and width marker seems unreliable since it is a PTM. The authors should use general tubulin staining to estimate centriole dimensions, or at least establish that acetylated tubulin correlates well with the overall tubulin signal in all mutants.

We have added two supplementary data figures (Fig 4 – supp 3 and Fig 4 – supp 4) in which we co-stain control and mutant centrioles with alpha-tubulin. We found that acetylated tubulin marked mutant centrioles well and as alpha-tubulin length increased, acetylated tubulin length also increased.

(9) Presence and absence of various centriolar proteins. These analyses lack a clear reference for the precise centriole elongation stage. This is particularly problematic for proteins that are recruited at specific later stages (such as inner scaffold proteins). The staining should be correlated with centriole length measurements, ideally using general tubulin staining.

As described for point 8, we have added two supplementary data figures in which we costain control and mutant centrioles with alpha-tubulin and found that acetylated tubulin also increases as overall tubulin length increases in all mutants. We note that inner scaffold proteins are absent in all our mutant centrioles at all stages of the cell and centriole cycle, as also previously reported for POC5 in Wang et al., 2017.

**Reviewer #2 (Recommendations For The Authors):**
Here's a list of points I think could be improved:- As the authors previously published, the centriole appears to have a smaller internal diameter than mature centrioles. Could the authors measure to see if the phenotype is identical? Is the centriole blocked in the bloom phase (Laporte et al. 2024)?

We have added an additional supplementary figure (Fig 4 – supp 5) to show that mutant centrioles have smaller diameters than mature centrioles, as we previously reported for the delta-tubulin and epsilon-tubulin mutant centrioles by EM. We thank the reviewers for the additional question of the bloom phase. Given the comparatively smaller number of centrioles we analyzed in this paper compared to Laporte et al (50 to 80 centrioles per condition here, versus 800 centrioles in Laporte et al), it is difficult to definitively conclude whether there is a block in bloom phase. This would be an interesting area for future research.

- The images of the centrioles in EM are beautiful. Would it be possible to apply a symmetrisation on it to better see the centriolar structures? For example, is the A-C linker present?

We thank the reviewer for this excellent suggestion. Using centrioleJ, we find that the A-C linker is absent from mutant centrioles. The symmetrized images have been added to Fig 1 – Supplementary Fig 2, and additional discussion has been added to the text (line 143-144, line 368-374).

- How many EM images were taken? Did the centrioles have 100% A-microtubule only or sometimes with B-MT?

For TEM, we focused on centrioles that were positioned to give perfect cross-section images of the centriolar microtubules, and thus did not take images of off-angle or rotated centrioles. Given the difficulty of this experiment (centrioles are small structures within the cell, centrosomes are single-copy organelles, and off-angle centrioles were not imaged), we were lucky to image 3 centrioles that were in perfect cross-section – 2 for *Tedc1-/-* and 1 for *Tedc2-/-*. Our images indicate that these centrioles only have A-tubules (Fig 1 – Supp Fig2).

- In Figure 2 - it would be preferable to write TEDC2-flag or TEDC1-flag and not TEDC2/1.

We have made this change

- It seems that Figures 2C and D aren't cited, and some of the data in the supplemental data are not described in the main text.

We have replaced these supplements with new supplementary figures for TEDC1 and TEDC2 localization (Fig 2 – Supp 1).

- The signal in U-ExM with the anti-Flag antibody is heterogeneous. Did the authors test several anti-FLAG antibodies in U-ExM?

We tested several anti-Flag and anti-V5 antibodies for our analyses, and chose these because they have little background signal in all applications (Fig 2 – Supplementary Fig 1E-J). Other commercially available antibodies against these tags did exhibit non-specific signal.

- The AlphaFold prediction is difficult to interpret, the authors should provide more views and the PDB file.

We have added 2 additional views of the AlphaFold prediction in Fig 3 – Supp 1A.

- In general, but particularly for Figure 4: the length doesn't seem to be divided by the expansion factor, it is therefore difficult to compare with known EM dimensions. Can the authors correct the scale bars?

We have corrected the scale bars for all figures to account for the expansion factor.

- Concerning Gamma-tubulin that is "recruited to the lumen of centrioles by the inner scaffold, had localization defects in mutant centrioles. However, we were unable to reliably detect gamma-tubulin within the lumen of control or de novo-formed centrioles in S or G2-phase (Figure 4 - Supplement 1E), and thus were unable to test this hypothesis". In Laporte et al 2024, Gamma-tubulin arrives later than the inner scaffold and only on mature centrioles, so this result appears to be in line with previous observation. However, the authors should be able to detect a proximal signal under the microtubules of the procentriole, is this the case?

We agree that this is an exciting question. However, in our expansion microscopy staining, we frequently observe that gamma-tubulin surrounds centrioles, corresponding to its role in the pericentriolar material (PCM). In our hands, we find it difficult to distinguish between centriolar gamma-tubulin at the base of the A-tubule from gamma-tubulin within the PCM.

- In the signal elongation of SAS-6, STIL, CEP135, CPAP, and CEP44, would it be possible to quantify the length of these signals (with dimensions divided by the expansion factor for comparison with known TEM distances)?

We have quantified the lengths of SAS-6 and CEP135 in new Fig 4 – Supp 3 and Fig 4 – Supp 4.

- The authors observe that centrin is present, but only as a SFI1 dot-like localization (which is another protein that would be interesting to look at), and not an inner scaffold localization. Can the authors elaborate? These results suggest that the distal part is correctly formed with only a microtubule singlet.

We agree with the reviewer’s interpretation that the centriole distal tip is likely correctly formed with only singlet microtubules, as both distal centrin and CP110 are present. We have added this point to the discussion (line 415).

-The authors observe that CPAP is elongated, but CPAP has two locations, proximal and distal. Is it distal or proximal elongation? Is the proximal signal of CPAP longer than that of CEP44 in the mutants? The authors discuss that the elongation could come from overexpression of CPAP, but here it seems that the centriole is not overlong, just the structures around the cartwheel.

We thank the reviewer for this point. It is difficult for us to conclude whether the proximal or distal region is extended in the mutants, as our mutant centrioles lacks a visible separation between these two regions. It would be interesting to probe this question in the future by testing whether subdomains of CPAP may be differentially regulated in our mutants.

**Reviewer #3 (Recommendations For The Authors):**
It isn't apparent to me what was counted in Figure 1C. Were all centrioles (mother centrioles and procentrioles) counted? Where is the 40% in control cells coming from? Can this set of data be presented differently?

We apologize for the confusion. In this figure, all centrioles were counted. We have updated the figure legend for clarity. We performed this analysis in a similar way as in Wang et al., 2017 to better compare phenotypes.

Figure 2C. and the text lines 182-187: The ultrastructural characterization of TEDC1 and TEDC2 suffers from the low quality of the TEDC1 and TEDC2 signals obtained postexpansion. In comparison with robust low-resolution immunosignal, it appears that most of the signal cannot be recovered after expansion. Another sub-resolution imaging method to re-analyze TEDC1 and TEDC22 localization would be essential. The same concern applies to Figures 2 - Supplement 2 and 3. Also, Figure 2 - Supplement 2 and Supplement 3 do not seem to be cited.

We thank the reviewer for these recommendations. As also mentioned above, we used an alternative super-resolution approach, a Yokogawa CSU-W1 SoRA confocal scanner (resolution = 120 nm), and found that TEDC1 and TEDC2 localize to procentrioles and the proximal end of parental centrioles (Fig 2 – Supplementary Figure 1a, b). Second, we used a recently described expansion gel chemistry (Kong et al., Methods Mol Biol 2024) combined with Abberior Star red and orange secondary antibodies. This technique resulted in robust signal at centrosomes and in the cytoplasm and indicated that TEDC1 and TEDC2 localize near the centriole walls of procentrioles and the proximal region of parental centrioles, near CEP44 (Fig 2 – Supplementary Figure 1c, d). These stainings complement and support our initial observations (Fig 2C, D) and we have edited the text to reflect this (lines 157-163). We have also removed the supplementary figures that were uncited in the text.

TUBD1 and TUBE1 form a dimer and TEDC2 and TEDC1 can interact. Any speculation as to why TEDC2 does not pull down both TUBE1 and TUBD1?

We apologize for the confusion. TEDC2 does pull down both TUBE1 and TUBD1 (Fig 3D, pull-down, second column, Tedc2-V5-APEX2 rescuing the *Tedc2-/-* cells pulls down TUBD1, TUBE1, and TEDC1).

Figure 4A and B. The authors use acetylated tubulin to determine the length of procentrioles in the S and G2 phases. However, procentrioles are not acetylated on their distal ends in these cell phase phases (as the authors also mention further in the text). Why has alpha tubulin not been used since it works well in U-ExM? The average size of the control, G2 procentrioles, seems too small in Figure 4A and not consistent with other imaging data (for instance, in Figure 4 - Supplement 1 C, Cp110, and CPAP staining). There is no statistical analysis in F4A.

We have added two supplementary data figures (Fig 4 – supp 3 and Fig 4 – supp 4) in which we co-stain control and mutant centrioles with alpha-tubulin. We found that acetylated tubulin correlates well with overall tubulin signal in all mutants. We have added statistical analysis to the figure legend of Fig 4A.

Lines 260 - 262: "These results indicate that centrioles with singlet microtubules can elongate to the same length as controls, and therefore that triplet microtubules are not essential for regulating centriole length." It is hard to agree with this statement. Mutant procentrioles show aberrantly elongated proximal signals of several tested proteins. In addition, in lines 326 - 328, the authors state that "Together, these results indicate that centrioles lacking compound microtubules are unable to properly regulate the length of the proximal end."

We thank the reviewer and have clarified the statement to state that these results indicate that centrioles with singlet microtubules can elongate to the same overall length as control centrioles in G2 phase.

Line 353: The authors suggest that elongated procentriole structure in mitosis may represent intermediates in centriole disassembly. Another interpretation, more in line with the EM data from Wang et al., 2017, would be that these mutant procentrioles first additionally elongate before they disassemble in late mitosis. The aberrant intermediate structure concept would need further exploration. For instance, anti-alpha/beta-tubulin antibodies could be used to investigate centriole microtubules.

We apologize for the confusion and have edited this section for clarity (lines 341-343): “We conclude that in our mutant cells, centrioles elongate in early mitosis to form an aberrant intermediate structure, followed by fragmentation in late mitosis.”

References need to be included in lines 122, 277, 279.

We have added these references

Line 281: Add references PMID: 30559430 and PMID: 32526902.

We have added these references (lines 265-266).

Line 289: "Moreover, our results suggest that centriole glutamylation is a multistep process, in which long glutamate side chains are added later during centriole maturation." This does not seem like an original observation. For instance, see PMID: 32526902.

We have added this reference (lines 273-274).